# Storylines of UK drought based on the 2010-2012 event

Wilson C.H. Chan[1], Theodore G. Shepherd[1], Katie A. Smith[2], Geoff Darch[3], Nigel W. Arnell[1]

[1]Department of Meteorology, University of Reading, Reading, UK
[2]UK Centre for Ecology and Hydrology, Wallingford, UK
[3]Anglian Water, Peterborough, UK

*Correspondence to*: Wilson Chan (wilson.chan@pgr.reading.ac.uk)

**Abstract.** Spatially extensive multi-year hydrological droughts cause significant environmental stress. The UK is expected to remain vulnerable to future multi-year droughts under climate change. Existing approaches to quantify hydrological impacts of climate change often rely solely on GCM projections following different emission scenarios. This may miss out low-
probability events with significant impacts. As a means of exploring such events, physical climate storyline approaches aim to quantify physically coherent articulations of how observed events could hypothetically have unfolded in alternative ways. This study uses the 2010-2012 drought, the most recent period of severe hydrological drought in the UK, as a basis, and analyses storylines based on changes to 1) precondition severity, 2) temporal drought sequence, and 3) climate change. Evidence from multiple storylines shows that maximum intensity, mean deficit and duration of the 2010-2012 drought were highly influenced
by its meteorological preconditions prior to drought inception, particularly for northern catchments at shorter time scales. The influence of progressively drier preconditions reflects both spatial variation in drought preconditions and the role of physical catchment characteristics, particularly hydrogeology in the propagation of multi-year droughts. Two plausible storylines of an additional dry year with dry winter conditions repeated either before the observed drought or replacing the observed dramatic drought termination confirm the vulnerability of UK catchments to a "three dry winter" storyline. Applying the UKCP18
climate projections, we find that drought conditions worsen with global warming with a mitigation of drought conditions by wetter winters in northern catchments at high warming levels. Comparison of the storylines with a benchmark drought (1975-76) and a protracted multi-year drought (1989-93) shows that for each storyline (including the climate change storylines), drought conditions could have matched and exceeded those experienced during the past droughts at catchments across the UK, particularly for southern catchments. The construction of storylines based on observed events can complement existing
methods to stress test UK catchments against plausible unrealized droughts.

## 1 Introduction

Droughts incur significant impacts on the natural environment and across multiple sectors. Meteorological droughts — continuous periods of below-normal precipitation — propagate through the hydrological cycle and translate into hydrological droughts – extended periods of below normal river flow or groundwater levels – and soil moisture droughts (Van Loon 2015).
Drought propagation, conditioned by catchment properties, can result in hydrological droughts that are significantly longer in

duration, more intense and affect a larger area (Barker et al. 2016). Hydrological droughts threaten water resources availability and incur additional environmental and socio-economic consequences. The UK has experienced several periods of severe hydrological droughts since the 1950s, including the "benchmark" 1975-76 drought (Marsh et al. 2007). However, although intense, this drought was relatively short-lived and other events are more significant in locations where hydrological systems are sensitive to longer droughts. The 1989-93 and 2004-06 droughts and the more recent 2010-12 drought further raised awareness of the vulnerability of the UK to future multi-year droughts under climate change. Previous research has shown that past UK multi-year droughts were characterized by at least one winter with significant precipitation deficit, and significant uncertainties remain over the role of remote climate drivers and changes to atmospheric circulation under climate change (Parry et al. 2012; Folland et al. 2015), which would affect the probability of multi-year precipitation deficits A deeper understanding of the causal factors of multi-year droughts is a significant challenge for current and future water management.

Climate change is expected to impact global water resources through changes to the quantity, quality and timing of river flow and other hydrological processes (Arnell and Gosling 2013). National-scale climate change assessment for the UK point to a general reduction in annual river flow except for western Scotland, with higher certainty over a decrease in summer flows for southern England and lower agreement over changes in winter flows (Arnell 2011; Prudhomme et al. 2012; Christierson et al. 2012). A recent synthesis identified that significant uncertainty remains over the magnitude of seasonal flow changes, with lower agreement on changes in the autumn and spring (Garner et al. 2017). Comparing two generations of UKCP probabilistic projections at 10 UK catchments, Kay et al. (2020) found that low and average flows at the selected catchments are projected to decrease in most cases for the 2050s, although the magnitude of change for UKCP18 is smaller compared to the UKCP09 projections.  Specific studies focusing on droughts point to increased drought intensity and frequency with more significant changes beyond the 2050s (Burke et al. 2010; Rahiz and New 2012; Dobson et al. 2020). Studies diverge on changes to the frequency and impacts of long duration droughts with some suggesting more intermittent, shorter-duration droughts (e.g. Blenkinsop and Fowler 2007; Chun et al. 2013a) and others highlighting large parts of the UK, particularly southern England, as hotspots for future multi-year droughts (e.g. Prudhomme et al. 2014; Brunner and Tallaksen 2019). Using the UKCP09 climate projections and a gridded hydrological model, Rudd et al. (2019) further found that there is a high likelihood of coincident hydrological droughts occurring in the Thames and Severn basins and that both peak drought intensity and duration are projected to increase in southeastern England particularly in the far future (2070s).

A common characteristic of existing studies is that they have predominantly been GCM-driven. This approach is top-down in nature as its outcomes are constrained by projections from the selected GCMs following different emission scenarios (e.g. RCPs, SSPs etc.). Often, these studies result in wide uncertainty ranges and are presented and employed in decision-making via the ensemble mean (Smith et al. 2018; Shepherd 2019). When quantitative scenarios are used as input to climate models and subsequently in impact models, multiple sources of uncertainties cascade and total uncertainty increases through each step of the modelling chain, a phenomenon which has been dubbed the "cascade of uncertainty" (Wilby and Dessai 2010). GCM-

related uncertainty — i.e. uncertainty among projected impacts from different climate models — is regularly cited as the largest source of uncertainty. This relates to uncertainty in projections of circulation-related aspects of climate change (e.g. precipitation) over land (Shepherd 2014). Although studies often attempt to analyse as much of the cascade of uncertainty as possible, even the most comprehensive ones are unable to fully analyse all sources of uncertainty along the entire modelling chain (Smith et al. 2018). Recent studies have thus tended to consist of increasingly computationally demanding data

processing workflows and its outcomes often involve large amounts of data presented with wide uncertainty ranges, which is not conducive for decision-making (Løhre et al. 2019).

This drive to disseminate probabilistic information from GCM projections may fail to adequately consider the full range of possible futures and in particular, the risks associated with low likelihood, high-impact events (Sutton 2019). This is

particularly the case with events involving persistent low-frequency atmospheric circulation regimes, which climate models struggle to represent accurately (Simpson et al. 2018). Given the deep uncertainties involved, bottom-up approaches have emerged to consider a wider range of plausible futures and aim to use GCM projections following different emissions and socio-economic scenarios as complementary information rather than as the only line of evidence. Scenario-neutral approaches explore system sensitivity through exploratory simulations on a two-dimensional response surface (e.g. changes in temperature

and precipitation seasonality) encompassing a wide range of plausible outcomes (Prudhomme et al. 2010). Similarly, decision scaling seeks to link response surfaces with specific decisions to identify thresholds where the system becomes unreliable (Brown et al. 2012). However, these approaches are designed as an initial screening tool and more detailed analysis of selected futures identified on the response surface are still needed (Prudhomme et al. 2015). A known limitation is that it is difficult to consider more than two dimensions at a time and may require multiple response surfaces to consider additional variables.

These approaches can also be resource intensive as they cover sensitivity over large ranges regardless of plausibility or empirical experience. Recent research also highlighted additional uncertainty in the methods selected to populate the response surfaces (e.g. RCM-scaling, weather generator or seasonal scaling: Keller et al. 2018).

New approaches have also been proposed to address specific concerns raised by the water resources industry and conduct

exploratory experiments to identify ways in which high impact events may develop. Recent studies have advocated for the creation of "tales" (Hazeleger et al. 2015) or "storylines" (Shepherd et al. 2018) of extreme events. Storylines are defined as physically self-consistent unfoldings of past events and the plausible evolution of these events in a future climate (Shepherd et al. 2018). Event storylines can be constructed to characterize how high impact events could hypothetically unfold given different changes to their physical drivers in both present and future climate. The drivers and impacts of every drought event

vary significantly. Analyzing the spatial coherence of European hydrological droughts since the 1960s, Hannaford et al. (2011) concluded that every drought event had distinctive drought signatures. There is therefore merit in looking at individual droughts following an event storyline approach (as opposed to aggregating over many dissimilar events). An event storyline approach operates on the basis of the observed event and enables a "forensic investigation" describing the impacts from a wide range of

plausible changes to causal factors of the event (Lloyd and Shepherd 2020; Doblas-Reyes et al. 2021). This approach is specifically designed to consider plausible high impact events and strengthen risk awareness to avoid type II errors (i.e. missed warnings) (Shepherd 2019). Storylines thus need not have probabilities attached and place emphasis on specific drivers of extreme events and how the associated impacts of the event may change given changes in those drivers. The storyline approach also allows for the creation of downward counterfactuals to reimagine how events could have turned out worse given changes to its characteristics and drivers (such as timing and sequence) (Lin et al. 2020). Although storylines are deterministic, there is a logical rationale for the storyline approach based on the fundamental principles of probability theory, given the deep uncertainties in the circulation response of climate change and its representation in climate models (Shepherd 2021). Recent examples of event-based studies include case study analyses of six past droughts in East Anglia (Lister et al. 2018), analysis of anomalous European temperatures during winter 2010 (Cattiaux et al. 2010), retrospective comparison of the 2003 and 2015 European summer droughts (Laaha et al. 2016), and an in-depth investigation of the seasonal drivers of the 2018 European heatwave (Bastos et al. 2020).

In this study, we select the 2010-12 UK drought as a case study from which different counterfactual storylines (i.e. events that did not happen in reality) are constructed. The aims of this research are to:
- Analyse the drivers and development of the 2010-12 UK drought and the geographic variation in hydrological response across UK catchments
- Create a number of storylines representing alternative unfoldings of the 2010-12 drought event with changes to 1) precondition severity, 2) temporal drought sequence, and 3) climate change at different warming levels
- Quantify and compare characteristics of the observed event and its storylines with those of selected severe droughts in the past

## 2 Methods

### 2.1 Streamflow data and hierarchical clustering

In this study, we make use of the Low Flow Benchmark Network (LFBN) designated by the National River Flow Archive (NRFA). The LFBN comprises of UK catchments that are deemed suitable for the low flow analysis given their near-natural conditions (Harrigan et al. 2018). We select the 100 catchments within the LFBN that are located in England, Scotland and Wales and which overlap with the catchments selected in previous studies of droughts by Smith et al. (2019) and Barker et al. (2019) (Fig. S1). Daily observed river flow ($m^3$/s) and catchment properties were extracted for each catchment from the NRFA via the *rnrfa* R package (Vitolo et al. 2016). The Standardized Streamflow Index (SSI) is used in this study to characterize drought events (Vincente-Serrano et al. 2012). The SSI is calculated by accumulating streamflow over a baseline period across a user-defined $n$ number of months. Daily observed and simulated river flow for each catchment is aggregated into mean monthly river flow. A probability distribution function is fitted to the $n$-month(s) accumulated monthly river flow for each

calendar month and standardized by transformation to a standard normal distribution. In this study, the tweedie distribution is selected to fit the accumulated streamflow. Comparing a number of probability distribution functions, Svensson et al. (2017) concluded that the tweedie distribution is most suitable for calculating drought indicators in UK catchments. The use of SSI fitted using the tweedie distribution has previously been employed to characterize hydrological drought risk in the UK in

Barker et al. (2015; 2019) and Arnell et al. (2021). Agglomerative hierarchical clustering, a dendrogram-based clustering approach, was used to group catchments with similar drought response during the 2010/12 drought using the *TSclust* R package (Montero and Vilar 2014). Similar hydrographs of SSI accumulated over 6 months (SSI-6) are grouped using the Ward's minimum variance method, which aims to minimize total within-cluster variance (Ward 1963).

## 2.2 Storylines considered in this study

The storyline approach provides a flexible means to investigate counterfactuals and the impacts of climate change. Storylines created for the 2010-12 drought are physical climate event storylines which focus on plausible changes to the causal elements behind an event to represent different ways in which the event could have unfolded. In this study, storylines are created by statistical adjustments to meteorological drivers, historical climate analogues and climate change impact assessment using the UKCP18 climate projections. Table 1 shows the various storylines considered in this study and example research questions

that each storyline aims to address.

**Table 1: Storylines considered in this study and description of example research questions**

| Storyline | Explanation | Example research questions |
|---|---|---|
| **Precondition severity** | | |
| Drier preconditions (DP) | 3- and 6-months prior to 2010-12 drought altered by estimated return periods (10, 20, 50 and 100-years) | How sensitive is the drought to progressively drier preconditions? |
| **Temporal sequence** | | |
| Seasonal contributions (SC) | Winter and autumn within event replaced with daily climatological precipitation and temperature (1965-2015) | What were the seasonal contributions to the development and termination of the drought? |
| Dry year before (DB) | Replace 2009 with a dry year (2010) before the 2010-2012 drought | What if the 2010-12 drought was preceded or succeeded by another dry year with dry winter conditions (i.e. a third dry winter situation)? |
| Dry year after (DA) | Replace 2012 with a dry year (2010) after the 2010-2012 drought | |
| **Climate change** | | |
| UKCP18 regional projections (GW) | UKCP18 projections applied to all months at 4 warming levels | What would happen if the 2010-12 drought occurred in a warmer world? |

### 2.2.1 Precondition severity

Precondition storylines of varying severity are generated for 3- and 6-months preceding the 2010-12 drought to investigate the
sensitivity of the event to progressively drier preconditions. The formulation of these storylines follows similar methodologies
employed in previous studies to investigate drought sensitivity in German (e.g. Stoelze et al. 2014; 2020) and Swiss catchments
(e.g. Staudinger et al. 2015). The preconditions are altered based on estimating return periods in precipitation over the particular
months. Specified return periods (10, 20, 50 and 100-year) are estimated from annual average 3-month (October-December)
and 6-month (July-December) precipitation for each of the 100 catchments for 1900-2015, and fitted with the generalized
extreme value (GEV) distribution. Observed precipitation for the 3- and 6-months prior to the 2010-12 drought is then reduced
or increased to match the estimated average precipitation at each return period. The temporal variability is therefore unchanged
from the observed precipitation of the specified 3- or 6-months period. The influence of the perturbed preconditions is
characterized by the precondition persistence time which is calculated from the start of the perturbation until the influence of
the perturbation is no longer detected. This is calculated as the number of days needed for river flow at each catchment to
return to values close to the baseline simulation (<1%). The precondition persistence time is not indicative of the time taken
for catchments to entirely recover from drought to non-drought conditions but is instead indicative of how long the influence
of precondition perturbations lasts for each catchment. This is consistent with indices used in Staudinger and Seibert (2014)
and Stoelze et al. (2020) to assess the influence on streamflow and groundwater recharge following perturbations applied to
initial conditions.


A consideration during the creation of these storylines is whether the perturbations violate the correlation structures between
PET and precipitation. Perturbing the precipitation prior to the observed drought independently of PET is plausible as observed
precipitation and PET for the period 1965-2015 exhibit no correlation except for a weak negative correlation in spring and
early summer (Supplementary Figure S2). Figure 1 shows that the resulting monthly precipitation after precipitation is reduced
are not outliers in the observed relationship between precipitation and PET. The creation of event storylines to understand the
role of preconditions in hydrological extremes in locations other than the UK may have to consider potential changes to the
correlation structures if a strong correlation between observed precipitation and PET (or other variables of interest) is found.
Additionally, the precondition perturbations do not violate existing autocorrelation structures as autocorrelation among
successive monthly precipitation values is mostly not statistically significant apart from the short term and decays rapidly after
the first few months (also seen when considering performance of stochastic weather generators; Kilsby et al. 2007; Serinaldi
and Kilsby 2012; Chun et al. 2013b).

### 2.2.2 Temporal sequence

Two sets of storylines are created by altering the temporal sequence of precipitation and temperature of the 2010-12 drought
by retaining certain periods and altering others based on historical observations. Firstly, to investigate the relative importance

of individual seasons in the development of the drought, we create storylines of seasonal contributions by prescribing daily climatological average precipitation and temperature for winter 2010/11 and 2011/12 and for autumn 2010 and 2011 while retaining observed values for the rest of the time series. The difference between the storylines and the baseline is indicative of the individual contribution of winter/autumn.

Secondly, we create storylines using historical climate analogues. These storylines are inspired by the "historical climate" approach employed by Hydrological Outlook UK where 3-month probabilistic projections of river flow trajectories are produced using hydrological model simulations driven by ensemble sequences of precipitation and temperature sampled from the historical record combined with up-to-date observations (Prudhomme et al. 2017). We create storylines exploring the hydrological impacts of a "third dry winter" situation — three consecutive dry years that includes consecutive drier than
average winters (Spraggs et al. 2015). The "Dry year before" storyline replaces the year preceding (i.e. 2009) the drought with a dry year. The "Dry year after" storyline replaces the year succeeding the drought with a dry year (i.e. from March 2012 to 2013) to explore the plausible consequences if drought conditions were not terminated by anomalously wet conditions in spring 2012. Although climate model projections indicate changes to average temperature and precipitation in a future period — for example, drier summers and wetter winters, these changes do not necessarily occur concurrently and may not be true for all
years. Consecutive dry seasons are possible and the hydrological response to long dry sequences merits further investigation. Past multi-year droughts have been shown to include at least one dry winter (Folland et al. 2015). Successive dry winters are shown to have caused significant reduction in river flows and reservoir storage in both observations and river flow reconstructions (Watts et al. 2012; Spraggs et al. 2015; Barker et al. 2019). Quantifying historical transition probabilities of consecutive dry half-years in England and Wales, Wilby et al. (2015) found that the longest consecutive dry half years spanned
4 years (incl. 4 dry winters) and that even longer dry sequences are possible. Additionally, a "third dry winter" situation specific to the 2010-2012 drought is a plausible case to investigate given widespread concerns during late 2011 and early 2012 when multiple water companies issued water use restrictions and applied for drought permits in anticipation of further depletion in water resources over 2012 based on the prevailing atmospheric conditions (Bell et al. 2013; Marsh et al. 2013).


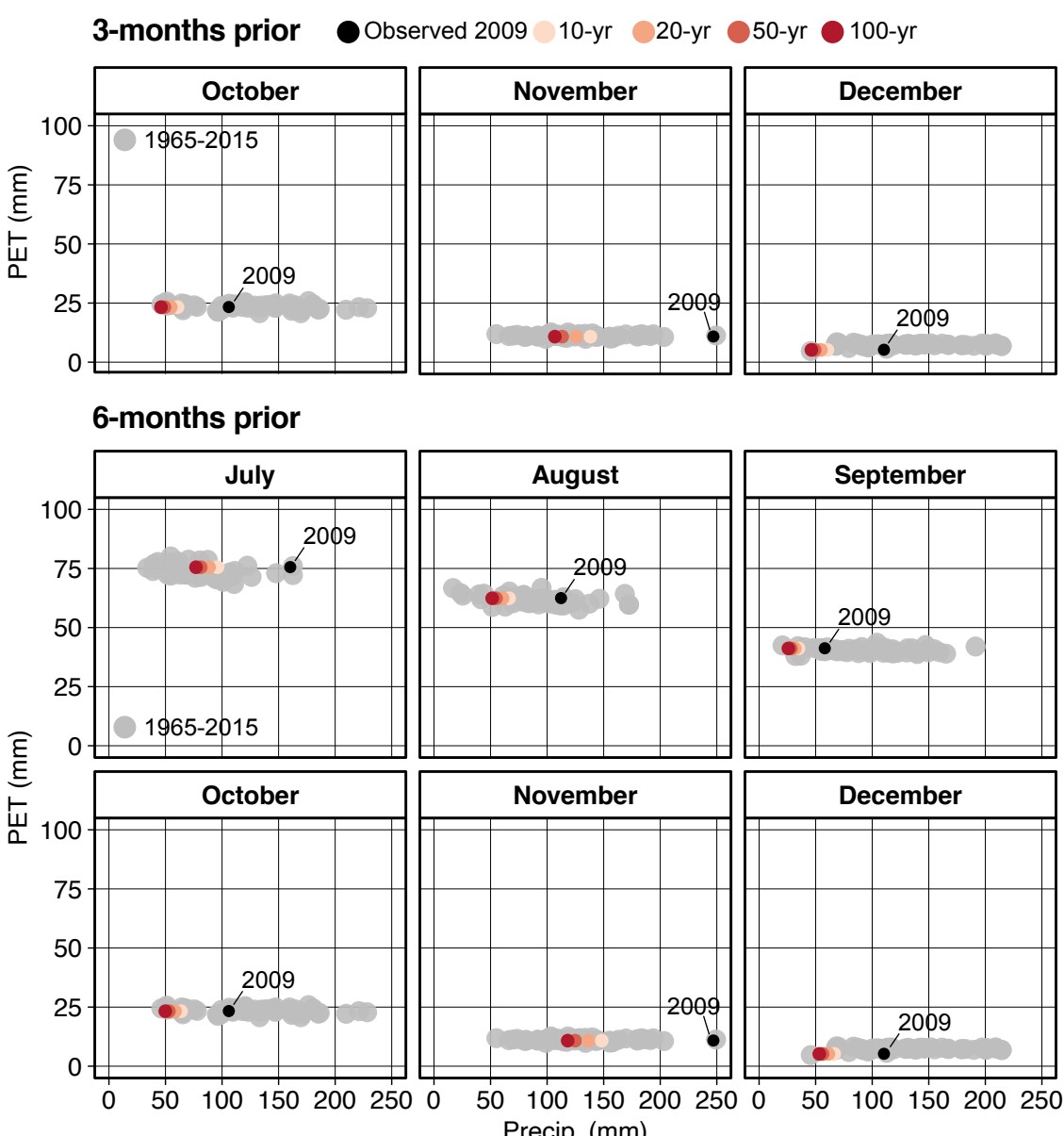

**Figure 1:** October to December (top) and July to December (bottom) monthly precipitation and PET (1965-2015) at an example catchment in southern England. The black circle indicates the observed value in 2009 and the coloured circles indicate the value after precipitation 3- and 6-months prior to the 2010-12 drought is reduced at four return periods.

### 2.2.3 Climate change

The UKCP18 12-member HadRM3 Perturbed Parameter Ensemble (PPE) regional climate projections at 12km resolution are used. The 12-member PPE is based on the HadGEM3 GCM and represents a plausible range of the climate model parameter space (Rostron et al. 2020). The regional projections are selected as they provide spatially coherent projections, important

given the spatial characteristics of droughts. A time-sampling approach (James et al. 2017) was used to select the 10-year time period starting from the year each ensemble member reaches conditions equivalent to four global warming levels (1.5, 2, 3 and 4°C) relative to 1981-2010. The delta change method is used to apply the projections (Anandhi et al. 2011). Monthly change factors for precipitation (%) and temperature (°C) are generated from comparing projections for a baseline period (1981-2010) to projections of the designated 10-year future periods. Change factors are generated for each river basin region designated by UKCP18. Change factors are applied either additively (for temperature) or multiplicatively (for precipitation) to the baseline temperature and precipitation for each selected catchment according to the river basin region the catchment is located in. The delta change method is widely and consistently employed in studies projecting the impacts of climate change across UK catchments (e.g. Arnell et al. 2003; Wilby and Harris 2006; Kay et al. 2020). In its standard form, this method retains the historical variability of the observations and changes in dry/wet spell lengths are not considered. Variations of the delta change approach has been applied to calculate percentile or quantile-based change factors to represent different magnitudes of relative changes in wet and dry days and short-duration rainfall intensity (e.g. Anandhi et al. 2011; Willems and Vrac 2011; Ntegeka et al. 2014).

Alternative statistical bias correction approaches such as quantile mapping exist to correct for different biases, but all techniques share the assumption that the biases corrected for and the bias correction technique itself remain valid for future time periods. While more complex statistical downscaling may be useful when assessing general changes in the impacts of climate change on hydrological variables, it is much more challenging to use bias corrected climate model data to search for similar analogues to observed events. Validating the plausibility of future events is inherently challenging, especially with concerns and uncertainty over the realism of climate model simulations for persistent circulation extremes and how atmospheric circulation patterns will change under climate change (Shepherd 2014). Previous studies have shown that climate models tend to underestimate drought persistence (important for multi-year droughts like the 2010-12 drought) and where multi-year droughts are simulated, the driving mechanisms vary between individual events (Ault et al. 2014; Moon et al. 2018) hence making it difficult to validate in relation to the observed 2010-12 event. The storylines of climate change aim to place the 2010-12 drought in a future climate instead of generalizing across dissimilar drought events. The underlying philosophy is that for such singular events, climate change (at least over the time frame of interest here) is a relatively small perturbation compared to natural variability, and so perturbing an observed event is preferable, in terms of physical realism, compared to making a large and inevitably aggregated bias correction to climate model projection. The delta change method is useful in this respect as by retaining the observed drought sequence, the meteorological conditions driving the observed multi-year drought remain consistent and plausible. This is an assumption that sacrifices the ability to generalize all extreme events but instead focuses on the specificity, and does not incur the uncertainties involved with bias correction and downscaling, which are significant contributors to the cascade of uncertainty. It also increases realism and familiarity with stakeholders and enable more interpretable comparison with the other storylines in the study which are also created based on altering the observed time series of the drought.

## 2.3 Drought characteristics

Maximum drought intensity, mean drought deficit and drought duration (months) are extracted using the SSI (Table 2). Simulated river flow accumulated over 6, 12 and 24 months are translated into the SSI for the calculation of drought characteristics. The parameter values for fitting the tweedie distribution are retained from the baseline and used to fit the distribution for each storyline. The same drought characteristics were used in Barker et al. (2019) to characterize historic droughts for the same set of UK catchments.

**Table 2: Drought characteristics considered in this study and their derivation method**

| Drought characteristic | Method |
|---|---|
| Drought event | Periods of negative SSI with at least one month reaching severe drought (SSI < -1.5). Catchments without a single month of severe drought are regarded as not under drought conditions. |
| Drought duration | Total number of months across all periods of identified drought conditions within event time frame |
| Mean deficit | Sum of all SSI/SPI values within periods of drought conditions (accumulated deficit) divided by drought duration |
| Max. intensity | Minimum SSI/SPI value across all identified periods of drought conditions within the event time frame |

## 2.4 Hydrological modelling and parameter uncertainty

The GR4J hydrological model is used to simulate the river flow for the baseline and storylines. GR4J is a daily lumped, bucket-type hydrological model with four model parameters available for calibration (Perrin et al. 2003). GR4J is driven by catchment-averaged daily precipitation (CEH-GEAR dataset; Robinson et al. 2020) and potential evapotranspiration (PET). PET is estimated using the temperature-based McGuinness-Bordne equation calculated from daily mean temperature (CEH CHESS dataset) with parameters tuned specifically for the UK (Tanguy et al. 2018).

Parameter uncertainty for GR4J for the selected catchments has previously been assessed by Smith et al. (2019). In Smith et al. (2019), GR4J was calibrated at the selected catchments using a multi-objective Latin hypercube sampling (LHS) strategy. 500,000 parameter sets for each catchment were produced and ranked at each catchment based on model performance assessed using multiple evaluation metrics (Nash-Sutcliffe efficiency (NSE), absolute percent bias (PBIAS), mean absolute percent error (MAPE), NSE on logarithmic flows (logNSE), absolute percent bias in Q95 and absolute percent error in 30-days mean annual minimum flows). The top 500 parameter sets (LHS500) were subsequently used to reconstruct historic river flows and Smith et al. (2019) demonstrated that they were able to reproduce periods of key historic droughts and their characteristics (timing and magnitude). As the original LHS500 ranking was done based on model performance over a long baseline period, we conduct a differential split-sample experiment to re-rank LHS500. For each catchment, the 10 driest years were selected based on mean annual precipitation (1965-2015). Model performance for each of the driest years was calculated using daily

observed and simulated river flow for four of the metrics in Smith et al. (2019): NSE, logNSE, MAPE and PBIAS. The metrics selected are unweighted as high flows (NSE), timing of flows (logNSE), flow variability (MAPE) and overall water balance (PBIAS) should be considered equally important for river flows during the driest years. For each catchment, the parameter sets are then ranked from best to worst for each metric and given a score (1 to 500, where a higher score implies worse performance). Finally, we re-rank LHS500 based on the total score — the sum of the scores for each parameter set for each metric. Retaining the new parameter set ranking, the performance metrics are re-calculated for each catchment, first for the ten wettest years and again for all years. By doing so, we investigate how model performance and parameter rankings change under different conditions.

Model performance is comparable between the new ranking (Dry rank) and the original rank (LHS500) (Fig.2). NSE and logNSE values using the top ranked parameter set in the Dry rank and show high values across most catchments. (Supplementary Figure S3). Notable outliers with relatively poorer model performance were fast-responding catchments in northern Scotland. These catchments, as identified in Smith et al. (2019), have "flashy" river regimes that are difficult to capture with possible influences from snowmelt that are not incorporated in GR4J. The split-sample experiment indicates that optimizing the parameter ranking from LHS500 based on dry conditions does not result in significant differences although for some catchments, the top parameter set in the Dry rank results in marginally better performance when only the driest years are considered. The top ranked parameter set in the original LHS500 ranking remains unchanged in the Dry rank for 17 out of the 100 catchments. For the majority of catchments (54 out of 100), the top parameter set in the new Dry rank is within the top 10 of the original LHS500 rankings. For the remaining catchments, the top parameter set in the new Dry rank are all found in the top 100 of the original LHS500 rankings (Supplementary Figure S3c). For the simulation of the baseline 2010-12 drought and its storylines, we make use the top-ranked parameter set from the Dry rank. Simulated river flows during the 2010-12 drought using the top ranked parameter set in the Dry rank is capable of reproducing periods of low river flows across the catchments during this period (Supplementary Figure S4).

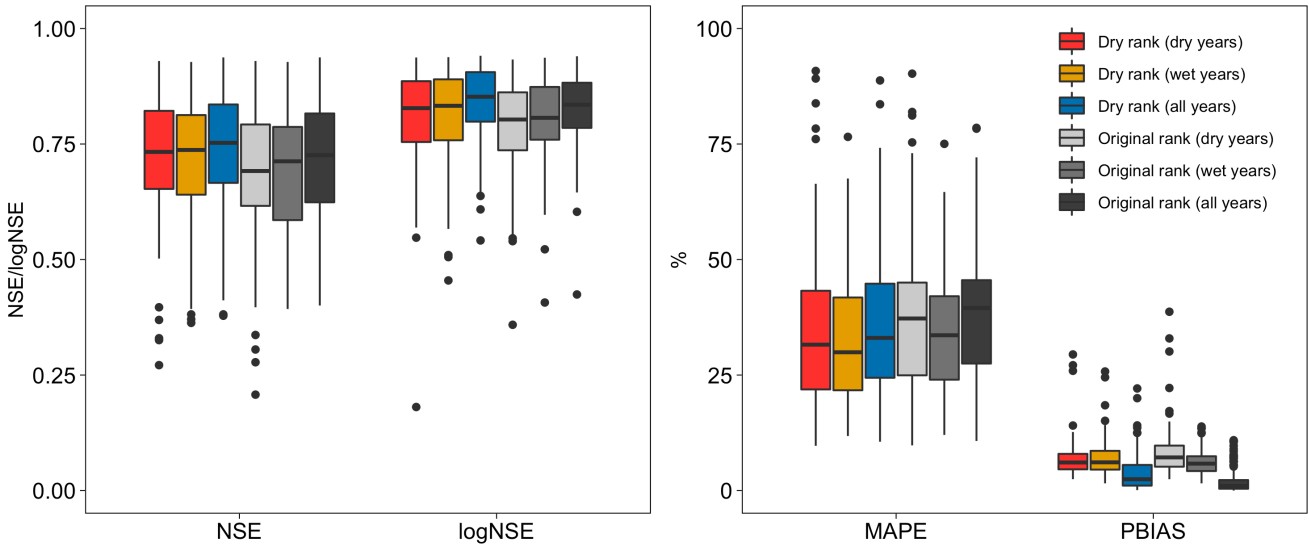

**Figure 2: Model performance of the top ranked parameter set across the selected catchments between parameter sets ranked based on the 10 driest years (Dry rank) and the original LHS500 rank (Original rank). Comparison is made for the top ranked parameter set in either the Dry rank or the Original rank when model performance metrics are calculated for the 10 driest years, 10 wettest years, and all years.**

## 3 Results

### 3.1 Anatomy of the 2010-12 drought

The 2010-12 drought was characterized by persistent blocked weather patterns over the UK from a northward shift of the jet stream over 2010 and 2011. Precipitation deficits were concentrated in winter, an important period when aquifer replenishment and reservoir re-fills normally occur (Kendon et al. 2013). The drought was notable for its dramatic termination when anomalously wet conditions occurred over spring and summer 2012, leading to a drought termination rate that was almost four

times quicker than other droughts in the observed record (Parry et al. 2013; 2016). The event ranks within the top ten most significant multi-year droughts in the English Lowlands for the past 100 years (Kendon et al. 2013). Drought orders were used by multiple water companies to supplement reservoir stocks, and temporary hosepipe and water use bans affecting over 20 million customers were ordered in early 2012, in anticipation of continued drought stress, prior to its abrupt termination (Environment Agency 2012; Kendon et al. 2013). The 2010-12 drought also incurred over £400 million in agricultural losses

and impacts to industrial activities from water use restrictions (Rey et al. 2017).

Figure 3 shows mean SSI accumulated over six months (SSI-6) across the event. The most severe conditions were experienced in southern England although the majority of catchments also experienced mild to severe drought conditions for a number of months between 2010 and 2012. Yearly autumn and winter precipitation and temperature anomalies (relative to 1965-2015)

showed that precipitation during winters 2009/10, 2010/11 and 2011/12 were all below average, confirming the importance of consecutive dry winter conditions in the development of the 2010-12 drought (Fig 3b). Autumn and winter are presented here as they represent crucial seasons when water resources are usually recharged. The exceptionally cold and dry conditions during winter 2009/10 were the precursor to the beginning of the drought where precipitation was significantly below average with blocked weather patterns across the western UK. The further northward shift of the jet stream in 2010 and across 2011 led to

the development of a significant NW/SE precipitation gradient with normal to above normal precipitation in the north and continued drier than average conditions in the south (Kendon et al. 2013).

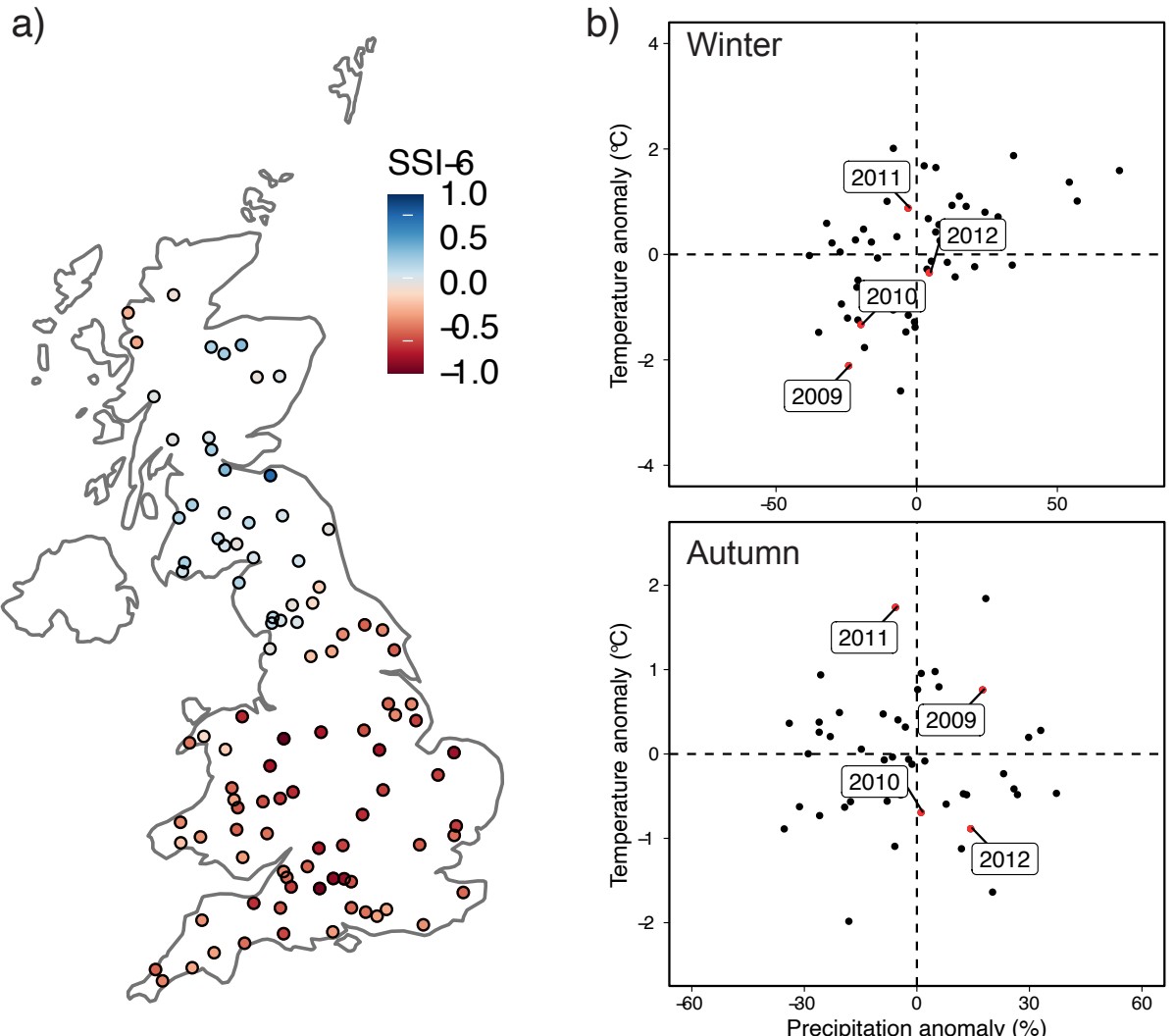

**Figure 3: a) Mean SSI-6 values between Jan 2010 and Mar 2012 b) Yearly winter and autumn precipitation and temperature**
**anomalies (relative to 1965-2015) averaged over the 100 selected catchments. 2009, 2010, 2011 and 2012 are shown by the red dots and the rest of the years are shown by the black dots.**

We use hierarchical clustering based on SSI-6 time series between January 2010 and March 2012 to identify groups of catchments with similar drought response (Fig.4). Cluster numbers between 2-10 were tested; five clusters are chosen as an appropriate number as this provides a clear distinction between hydrogeological units across southern England. The diversity of hydrological response to droughts in groundwater-dominated catchments in southern England has previously been shown in Merchant and Bloomfield (2018), and differences in hydrological drought response among catchments in this region should be considered. The use of five clusters also divides the northern catchments into east and west Scotland and distinguishes catchments in eastern Scotland where the influence of snowmelt processes may be more prevalent (also catchments with relatively poorer model performance). We select SSI-6 to delineate clusters instead of longer accumulated periods as it allows for a greater separation of catchments based on a larger variation in short-term drought response. SSI calculated with longer accumulation periods leads to a grouping of the hydrological response where only two clusters can be qualitatively identified. Subsequent storyline analyses will employ SSI-6, 12 and 24 in order to consider the role of catchment memory.

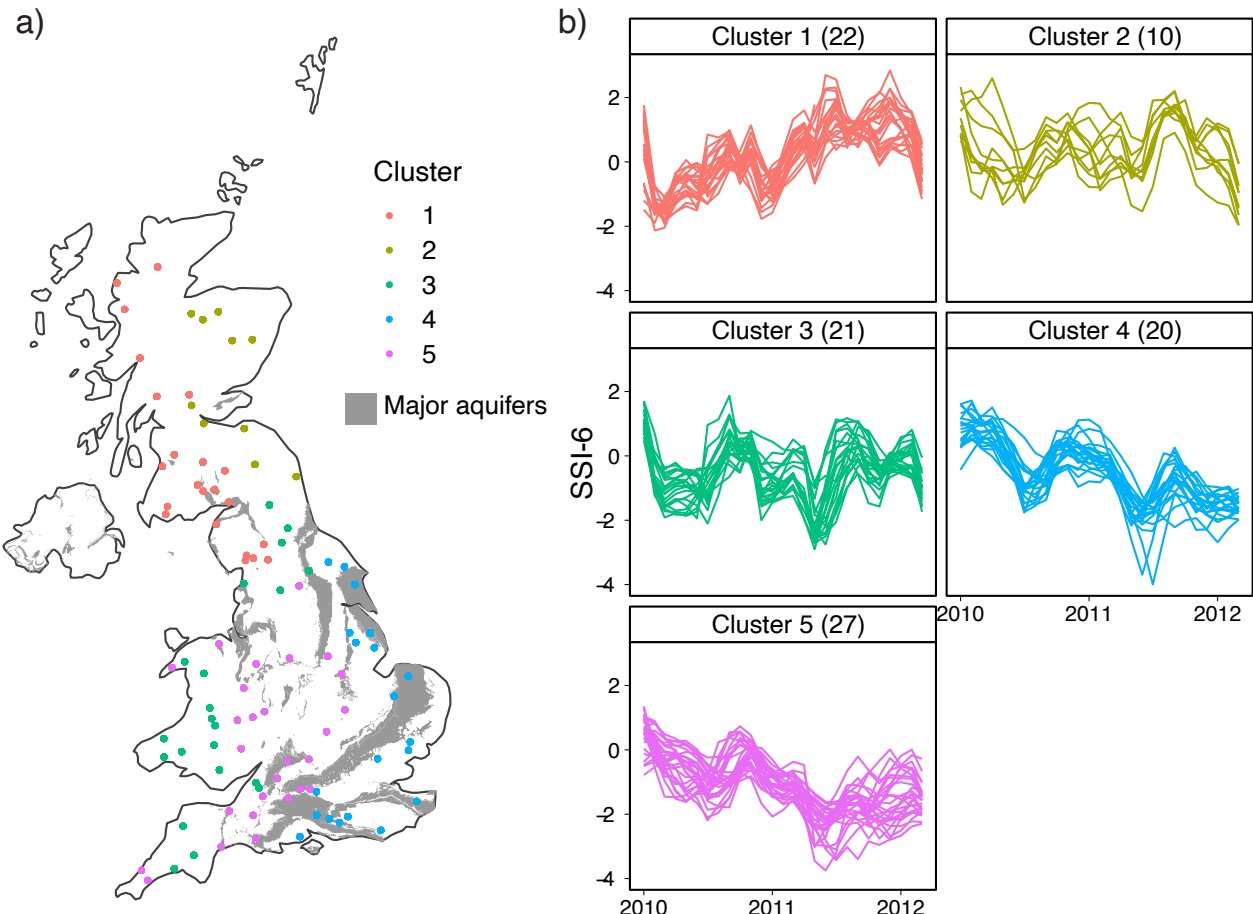


**Figure 4: Hierarchical clustering of SSI-6 during the 2010-2012 drought event. a) Spatial variation in the five identified clusters. b) SSI-6 between Jan 2010 and Dec 2012 for the catchments in each cluster.**

Initial streamflow response was relatively uniform in response to precipitation deficit in early 2010 for all clusters with moderate to severe drought conditions (SSI < -1.5). Catchments in Clusters 4 and 5 (southern and southeast England) were the

most significantly affected, with severe drought conditions developing as a result of the influence of a second consecutive dry winter. The majority of catchments in Cluster 4 are underlain by chalk aquifers and are slow-responding catchments with significant groundwater storage. Catchments in Cluster 3 (southwest England) saw severe drought conditions develop over late 2010 and 2011 but the impacts did not persist as long and were not as severe as Clusters 4 and 5. Conversely, catchments in Clusters 1 and 2 (west and east Scotland) were less affected. Although mean SSI-6 over 2010 and 2012 was not particularly

severe, the SSI-6 time series for these catchments show mild to severe drought conditions in initial response to precipitation deficit over winter 2009/10, after which streamflow recovered and did not descend to further severe drought conditions. Catchments in Cluster 2 were the least affected, with streamflow not reaching severe drought conditions at any point.

## 3.2 Storylines of seasonal contributions

Storylines of seasonal contributions reveal the relative importance of individual seasons in the development of the 2010-12
drought. Figure 5 shows cluster mean SSI-6 for the baseline and the two storylines of seasonal contributions (Figs S5 and S6
for SSI-12 and 24). The storylines confirm the importance of dry winters in the development of multi-year droughts. Drier
than average winters in 2010/11 and 2011/12 were a major determinant of the severe drought conditions observed across all
clusters apart from Cluster 1 for winter 2011/12. Baseline drought conditions across 2011, particularly for catchments in
southern England (Clusters 4 and 5), can be attributed to an abnormally cold start to winter 2010/11 and low precipitation.
Drier than average winter 2011/12 prolonged the dramatic drought termination for all clusters apart from Cluster 1. For Cluster
1, winter 2011/12 were wetter than average and the replacement of winter 2011/12 with climatology meant that catchments
could have experienced short-term minor drought conditions before recovery from wet conditions in 2012.

In addition to dry winter conditions, wetter than average autumn 2010 prevented catchments in all clusters from an earlier
drought inception and more intense drought conditions apart from Cluster 5. For Cluster 5, autumn 2010 was drier than average
which accelerated drought inception for SSI-6 but the effects are less noticeable at longer accumulation periods. Conversely,
autumn 2011 was drier than average which exacerbated drought conditions across all clusters when combined with drier than
average winter conditions apart from Cluster 1 and 2. The most affected catchments in Cluster 4 and 5 would have begun
drought recovery earlier and dry winter conditions along would not have been enough to prolong drought conditions as seen
in the baseline. For Cluster 1 and 2, wetter than average autumn 2011 prevented recovered catchments from returning to mild
drought conditions, particularly when considering longer accumulation periods

In summary, the wetter (drier) than average autumn 2010 (2011) resulted in diverging effects for catchments in Cluster 1 and
2 compared to Cluster 3-5. Drier than average winter 2010/11 and 2011/12 worsened drought conditions. At the most affected
catchments, the effects of dry winters is most notable for SSI-24, highlighting the role of catchment storage in attenuating the
effects of dry winter conditions in drought development. Autumn conditions was a determinant of the timing of drought
inception while winter conditions were important in determining the drought's eventual length.

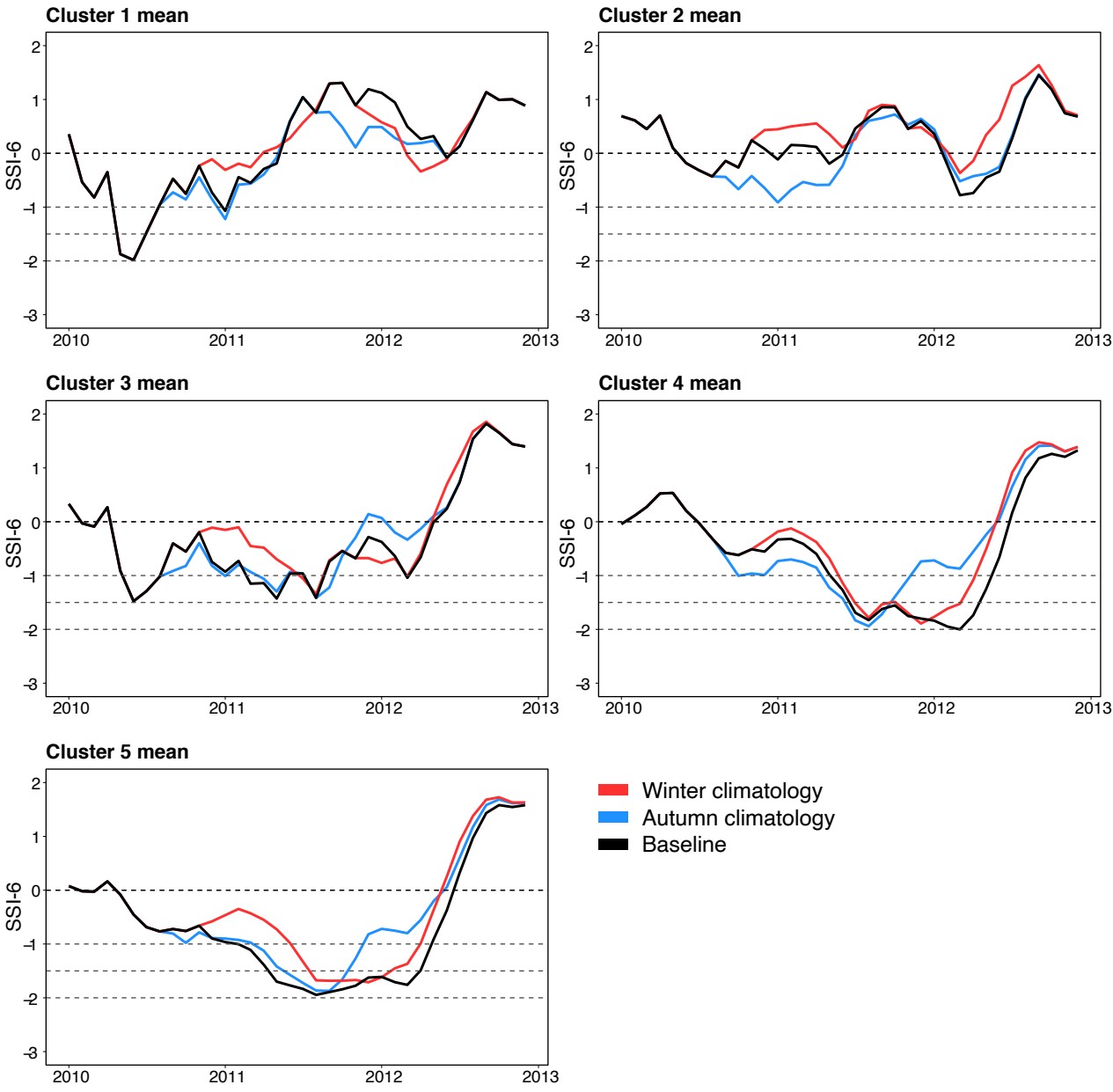

**Figure 5: Cluster mean SSI-6 for the storylines of seasonal contributions with winter 2010/2011 (red) and with autumn 2010 and 2011 (blue) replaced by daily climatological values. See supplementary figures S5 and S6 for the equivalent figure for SSI-12 and SSI-24.**

### 3.3 Storylines of precondition severity

Prescribing drier preconditions at varying severity for the 3- and 6-months prior to the 2010-12 drought reveals the influence of preconditions on the baseline event (Fig.6). As the aims of altering precondition severity are to investigate short-term catchment sensitivity to drier preconditions, only SSI-6 is used here. Drier preconditions lead to 12-month precipitation prior to the drought varying between 65-107% (48-90%) relative to the long-term average for the 3-months (6-months) precipitation reduction, with significantly greater deficit for catchments in Clusters 4 and 5 (Supplementary Fig.S7). Unsurprisingly, both

max. drought intensity and mean drought deficit increase in most cases for all clusters with an increase in precondition severity. The exception is Cluster 2 where changes in precondition precipitation with a 10- and 20- return periods lead to a reduction in drought intensity and deficit, meaning that the dryness observed 3- and 6-months precipitation prior to the 2010/12 event had a return period greater than 20 years for these catchments. Max. intensity and mean deficit are more sensitive for catchments in Clusters 1-4 compared with those in Cluster 5, especially for perturbations at shorter return periods. The difference between

the two precondition lengths is most notable for catchments in Clusters 1 and 2, where a 6-month precondition length results in much greater change in drought characteristics. Precondition length is less important for catchments in Clusters 5, indicating that the conditions that developed over the 3-months prior to 2010 (i.e. winter 2009/10) were already dry enough for the development of severe drought conditions, and only preconditions with longer return periods would result in significant differences to the eventual drought characteristics.


The influence of the most severe preconditions (i.e. 6-months, 100-yr return period) separates clusters into relatively fast-responding (Clusters 1-3) and slow-responding (Clusters 4 and 5) (Fig.6C). Drought conditions at fast-responding catchments are sensitive to the least severe preconditions with only a 10-yr return period. Conversely, change in max. intensity is relatively minimal for slow-responding catchments and is only notable with preconditions beyond a 20-year return period. Spatial

variation in the precondition persistence time differentiates catchments based on latitude with those in southern England showing the longest persistence time, coinciding with regions of major aquifers (Fig.7). Persistence time also differentiates latitudinal differences in catchment properties shown in Table 3 (Fig.7). There is a positive relationship between persistence time and both the baseflow index (BFI) and the proportion of arable/horticultural land. Higher values of the BFI are associated with more permeable catchments in the English lowlands. These catchments have high groundwater storage which contributes

to surface streamflow during drought and are more associated with agricultural/horticultural activities compared to impermeable catchments. Catchments with longer persistence times also tend to be larger in size and less steep. Additionally, climate properties show these catchments receive lower annual average precipitation and exhibit dry soil moisture for a larger proportion of time. This confirms that permeable lowland catchments are more vulnerable to long drought propagation with a lag (and lengthening) between meteorological and hydrological droughts. Drought response from drier preconditions shows

that catchment sensitivity reflects a combination of spatial characteristics of the drought and catchment properties, particularly the influence of hydrogeology.

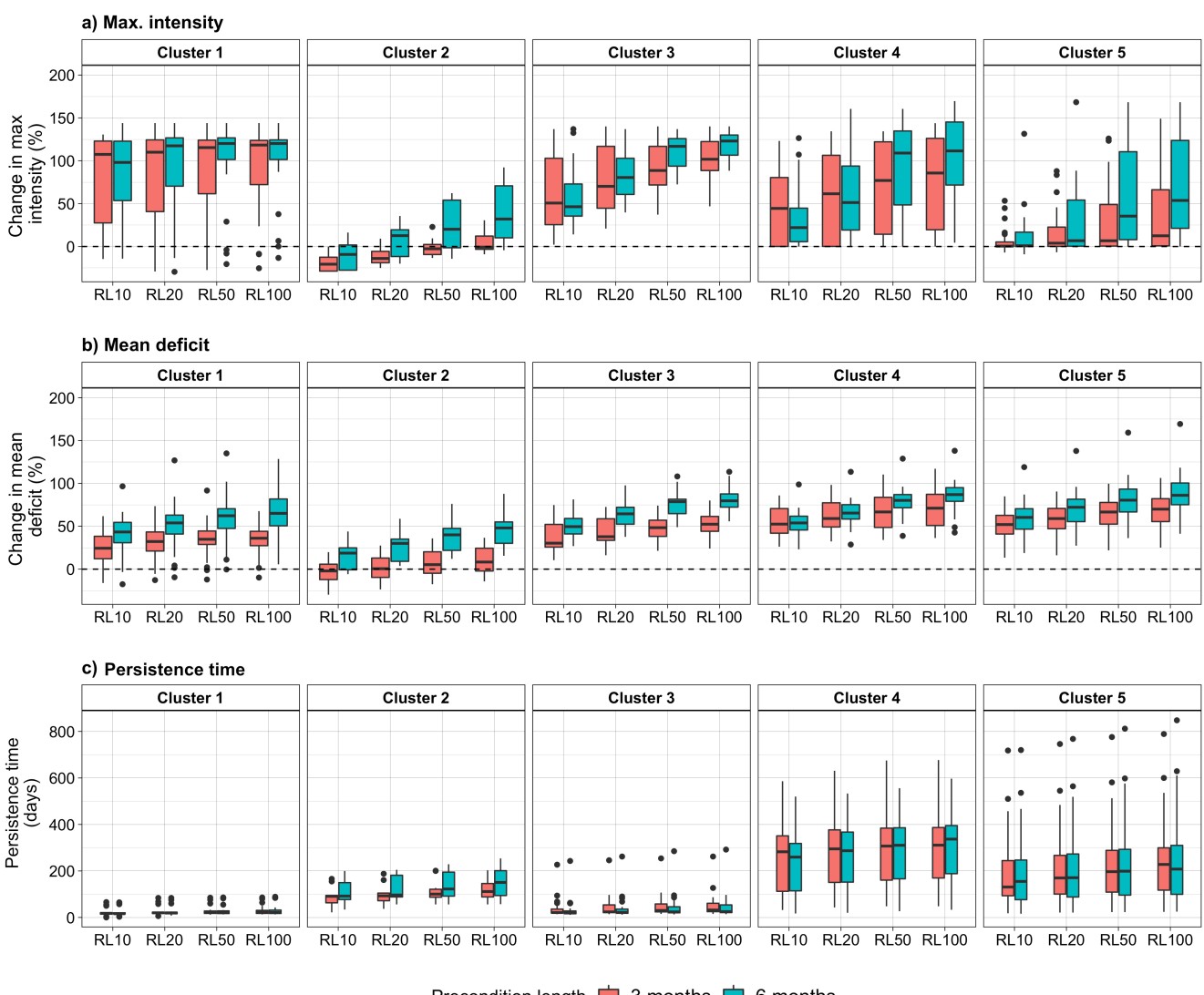

**Figure 6: Change in mean drought deficit (%), max. drought intensity (%) and persistence time (days) from the storylines of precondition severity at different return levels calculated from SSI-6.**



**Table 3: Description of selected catchment properties**

| Catchment properties | Description |
|---|---|
| Catchment area (km$^2$) | Total area of the catchment (km$^2$) |
| DPSBAR (m/km) – catchment steepness | Mean drainage path slope (DPSBAR) is an index for catchment steepness calculated as the mean inter-nodal slopes within a catchment. Higher values indicate steeper terrain and lower values flatter terrain. |
| PROPWET (%) | Proportion of time soils within a catchment are designated as being wet (i.e. higher values indicate wetter). PROPWET varies from <20% to >80% across the UK. |
| Proportion of horticultural/arable land (%) | Land use information derived from the Land Cover Map 2000 and the NRFA Land Cover Classes 2000 |
| BFI | Baseflow Index (BFI) is a measure of the proportion of river flow that derives from groundwater storage. Higher values indicate more permeable catchments with high groundwater contribution to river flow during dry periods. |
| SAAR 1961-1990 (mm) | Standardized Annual Average Rainfall (SAAR) over 1961-1990 30-year period |

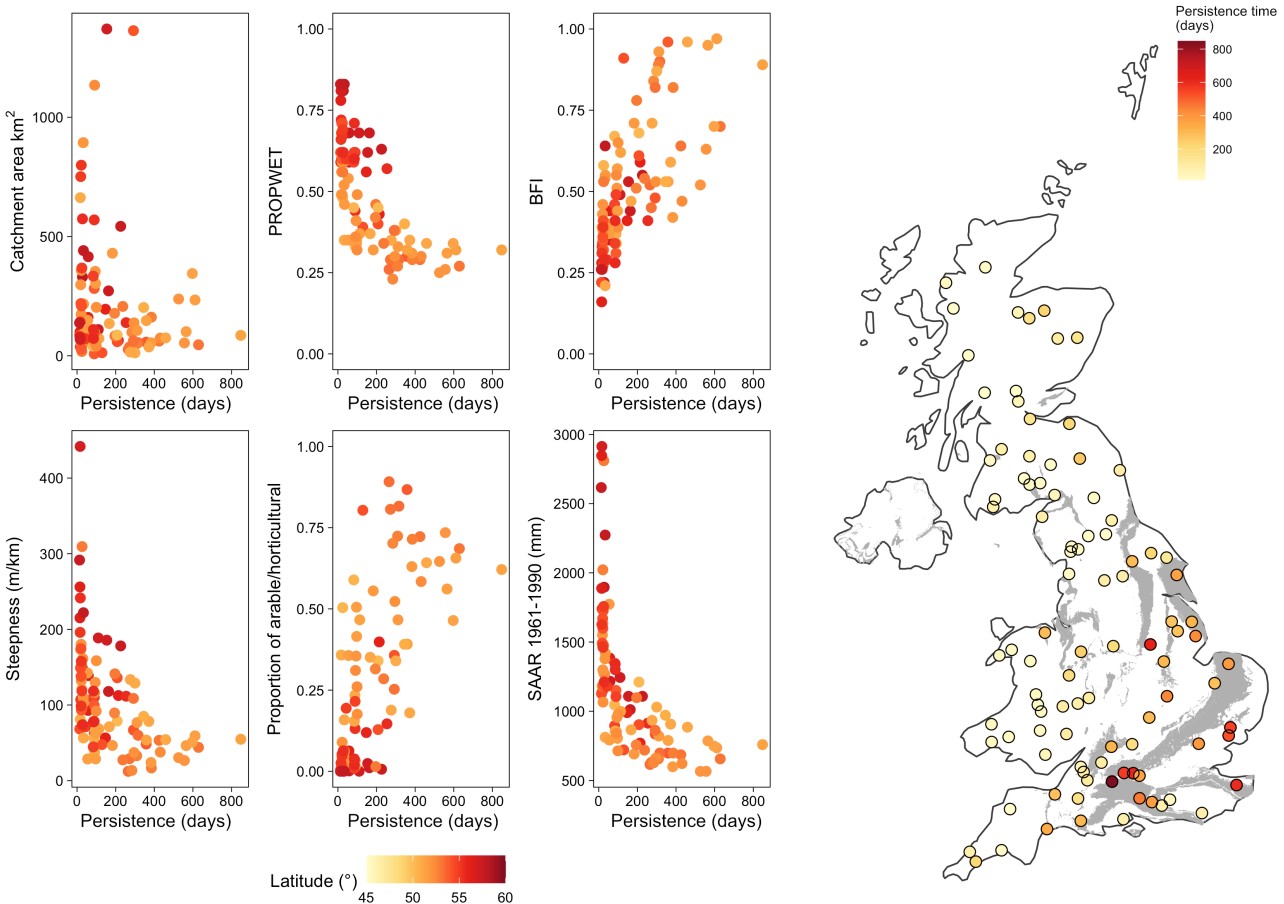

**Figure 7: a) Relationship between persistence time (days) of the 6-month precondition storyline (100-year return period) with selected catchment characteristics and b) Spatial variation of persistence time for the selected catchments.**


## 3.4 Storylines of temporal sequence

Altering the temporal drought sequence illustrates how much worse the 2010-12 drought could have been, given another dry year (Fig.8). The drought defined by SSI-6 is estimated to worsen for the "Dry year before" storyline for all clusters except for mean drought deficit for Cluster 4. The anomalous reduction in mean drought deficit for Cluster 4 at SSI-6 relates to an increase in drought duration that is greater than the increase in accumulated deficit and maximum intensity. For this storyline, change in drought characteristics are greatest for Clusters 1 and 3, with a larger increase with longer accumulation periods. This indicates that the addition of a dry year prior to the observed event increases the risk of abrupt and intense drought conditions in these catchments. Changes in drought conditions are significant enough and is noticeable at longer accumulation periods, despite the relatively fast precondition persistence times for catchments in these clusters. Conversely, change in max. intensity and mean deficit for catchments in Clusters 4 and 5 is notable only at longer accumulation periods. The larger change in drought characteristics for SSI-24 is particularly important for Clusters 4 and 5 as long accumulation periods are often used to assess drought at these slow-responding catchments with significant catchment storage.

Compared to the "Dry year before" storyline, the "Dry year after" storyline has a greater effect in the worst affected catchments in southern England. Without the dramatic drought termination in 2012, drought duration would have increased significantly for catchments in all clusters. Max. intensity and mean deficit are estimated to increase for all clusters, with the largest increase for Cluster 4 followed by Cluster 5 at all accumulation periods. This suggests that there is still considerable scope for even worse drought conditions to develop if dry conditions had persisted across 2012. The change in max. intensity is greatest for SSI-12 for all clusters except Cluster 5 while the magnitude of change in mean deficit increases with accumulation period and is greatest (smallest) for SSI-24 for Cluster 3-5 (Clusters 1-2). This indicates the importance of assessing drought conditions at multiple accumulation periods and highlights the importance of catchment and water resource memory. At Clusters 1-2, SSI-6 and 12 are useful to capture changes in drought conditions from the storylines but for Clusters 3-5, SSI-12 or longer are needed to fully assess the drought response.

Individual catchment examples again reflect the role of catchment response time in determining the catchment sensitivity to an additional dry winter. Figure 9 shows SSI-6 of nine catchments spanning the five clusters for the two storylines (Figs S5 and S6 for SSI-12 and 24). We can identify three categories of response. First are fast-responding catchments (e.g. 81002 – Cluster 1, 7001 – Cluster 2) that recover from both the "Dry year before" and "Dry year after" storylines quickly, with changes observable only for the perturbed year. Second are slow-responding catchments (e.g. 38026 and 42008 – Cluster 4) where streamflow response from the "Dry year before" storyline persists across 2010 but not significantly beyond 2011. Third are slow-responding catchments (e.g. 43014 and 39019 – Cluster 5) where streamflow response to the "Dry year before" storyline persists across 2010 and beyond into 2011. The "Dry year after" storyline also shows that even with continued dry conditions,

the meteorological conditions over 2013 would still have been wet enough to allow the most affected catchments to exit drought conditions.

470

In summary, the impacts of the "Dry year before" and "Dry year after" storylines vary spatially. The impacts of the "Dry year before" storyline is particularly severe for catchments in Clusters 1 and 3 although impacts remain apparent when considering catchment memory for Cluster 5. The impacts of the "Dry year after" storyline is particularly severe for Clusters 4 and 5, highlighting the role of catchment storage in slow drought propagation.

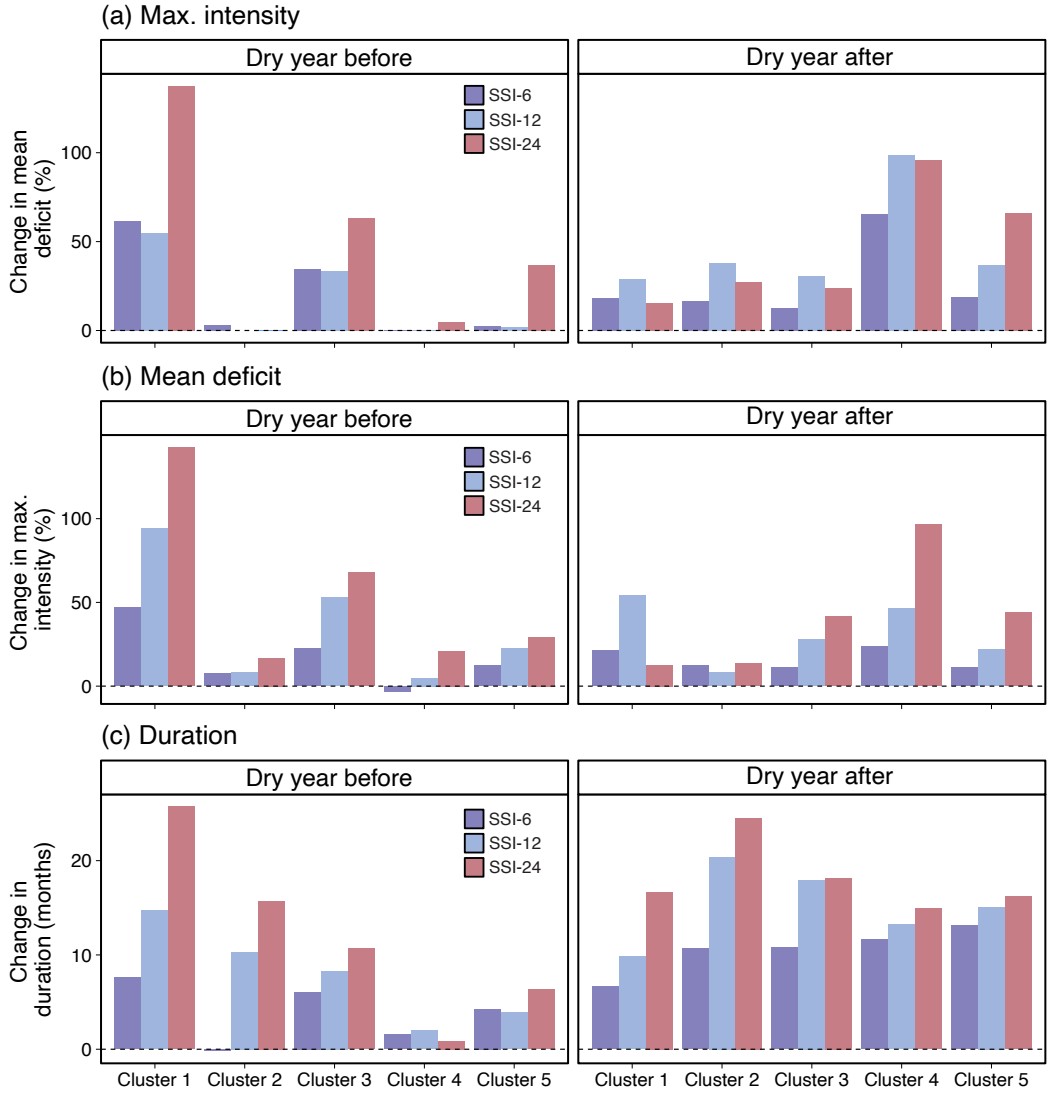

475

**Figure 8: Mean change in max. drought intensity (%), mean drought deficit (%) and duration (months) relative to the baseline for each cluster for either repetition of a dry year (2010) before (left) and after (right) the 2010-2012 drought.**

480

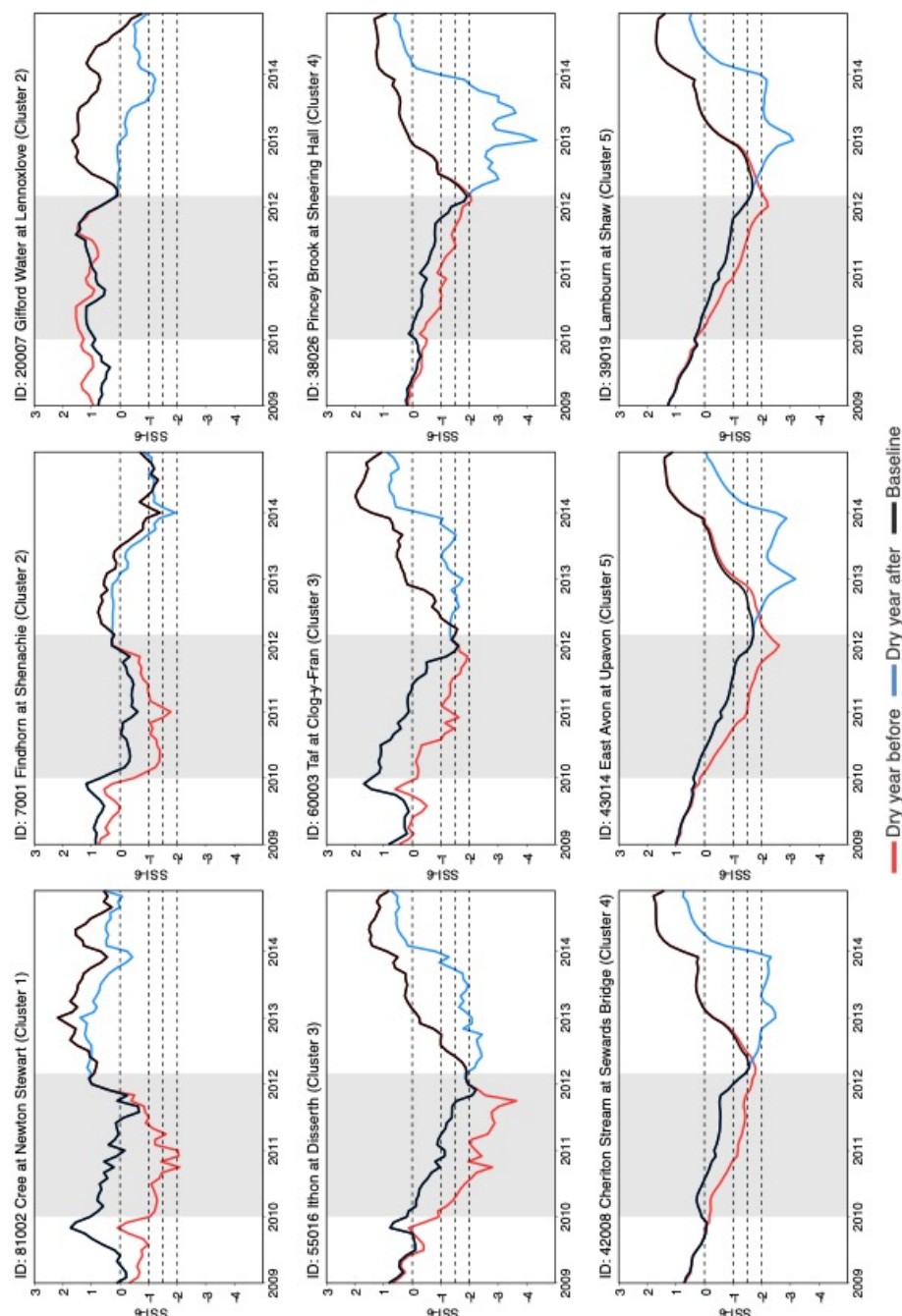

**Figure 9: Baseline (black) and simulated SSI-6 for a repetition of a dry year before (red) or after (blue) the 2010-2012 drought for nine example catchments spanning the five hydrograph clusters. The shaded region indicates the duration of the baseline 2010-2012 drought event (January 2010 to March 2012). See Figure 10 for the locations of the nine example catchments. See Figures S8 and S9 for SSI-12 and SSI-24**

## 3.5 Storylines of climate change

The UKCP18 regional projections were used to place the 2010-12 drought under future warming. The projections point towards, in general, wetter winters and drier summers with increasing temperature rise (Fig. 10). This climate change-induced change in seasonality of precipitation is particularly noticeable at 3°C and 4°C rise in temperature, with general agreement among the 12 regional projections over the sign of change. Projections also point to increased seasonality in temperature changes with greatest change in temperature in the summer, reaching 6°C higher relative to 1981-2010 in the summer of a 4°C world. Change in annual mean precipitation for each of the catchment clusters across the 12 regional projections are shown in Supplementary Fig.10.

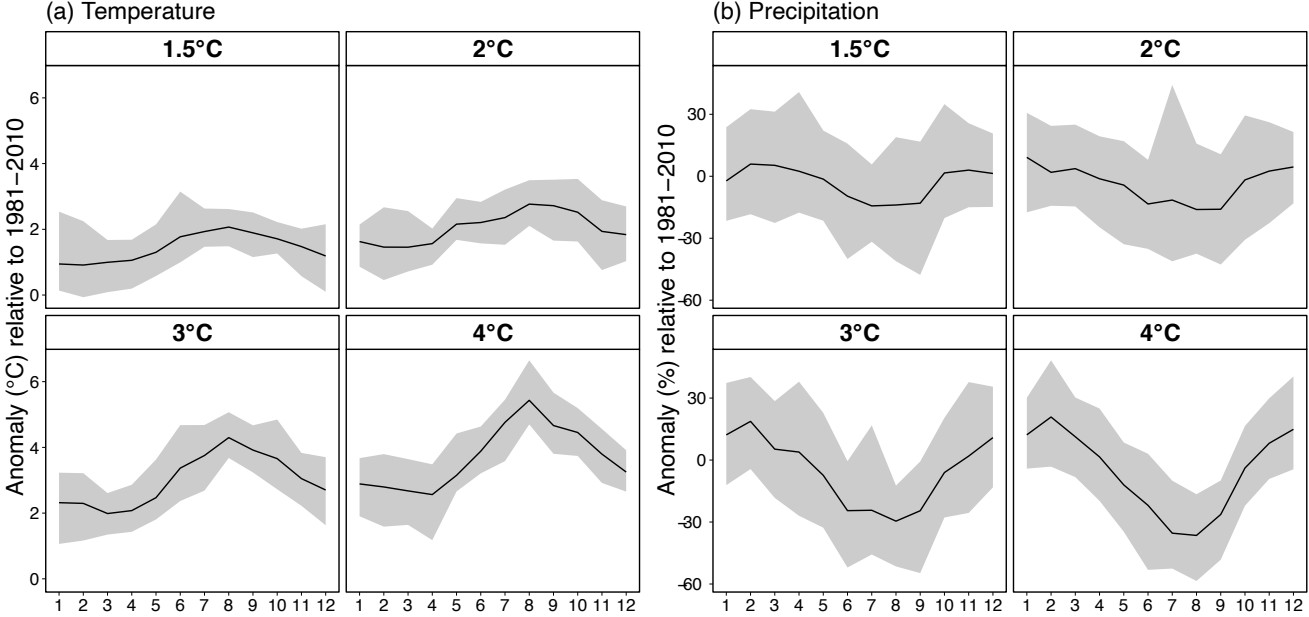

**Figure 10: Projected percentage change in monthly average a) temperature and b) precipitation relative to 1981-2010 from the UKCP18 regional projections at four warming levels averaged across the 100 selected catchments. The shaded region represents the maximum and minimum range of projected change amongst the 12 regional projections. The solid black line represents the ensemble mean.**

Under climate change, river flow across the 2010-12 drought is projected to decrease for the majority of catchments (Fig.11). In fast responding catchments (Clusters 1 and 2), winter river flows increase due to the projected increase in winter precipitation. In these catchments, the buffer effects of wetter winters compensate for increased evaporative demand from increased temperature. Mean discharge across the drought event for catchments in southern England and Wales is projected to decline substantially, with progressively larger declines at higher warming levels. River flow is projected to decrease in all seasons for even a 1.5°C rise in temperature with increasingly drier conditions at high warming levels, particularly for slow-

responding catchments in the south (Clusters 4 and 5). In these catchments, river flow is also projected to decrease progressively over the event timescale.

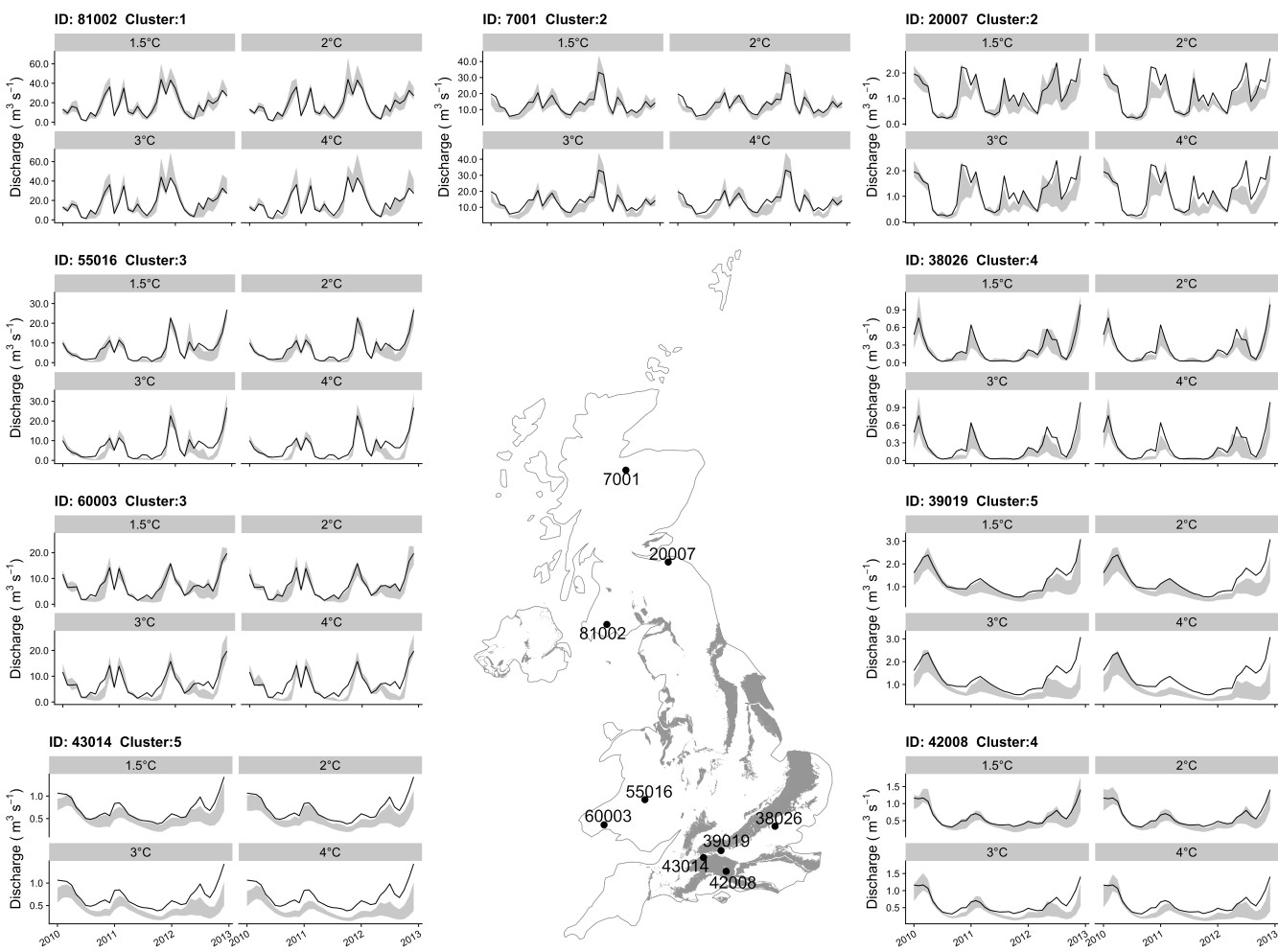

**Figure 11: Projected change in river discharge across 2010-2012 at four warming levels. Nine example catchments spanning the five hydrograph clusters are presented here. The solid line represents the baseline simulation, and the shaded region represents the uncertainty range of the 12 UKCP18 regional projections. Shaded regions on the map indicate the location of major aquifers.**

Given the observed drought sequence, conditions of the 2010-12 drought are projected to worsen with global warming across all clusters for all accumulation periods (Fig.12). Percentage change in drought characteristics relative to baseline for initial temperature rise (1.5°C and 2°C) is greater for Clusters 3-5 compared to Clusters 1 and 2. Beyond 2°C, drought characteristics are projected to worsen by a similar magnitude for all clusters except Cluster 1. The magnitude of change is larger at longer accumulation periods for Clusters 2-5. Although max. intensity and mean deficit are projected to increase with temperature

rise for Cluster 1, the increase in drought duration at 4°C is smaller compared to lower warming levels, indicating more intense
drought conditions with greater deficit despite a smaller increase in drought duration. For SSI-12 and 24, the magnitude of
change in drought characteristics is comparable between the warming levels without the clear progressive increase in drought
characteristics seen using SSI-6. This reflects the fast response times and limited catchment memory for these catchments
where drought conditions are better captured using short accumulation periods. Nonetheless, for SSI-12 and 24, mean drought
deficit at 3°C and 4°C and in max. intensity at 4°C is estimated to be smaller than the change projected for lower warming
levels. This anomalous behaviour could be attributed to wetter winters for northern Scotland, especially at high warming levels,
which provide wet interludes and mitigate drought conditions of the event.

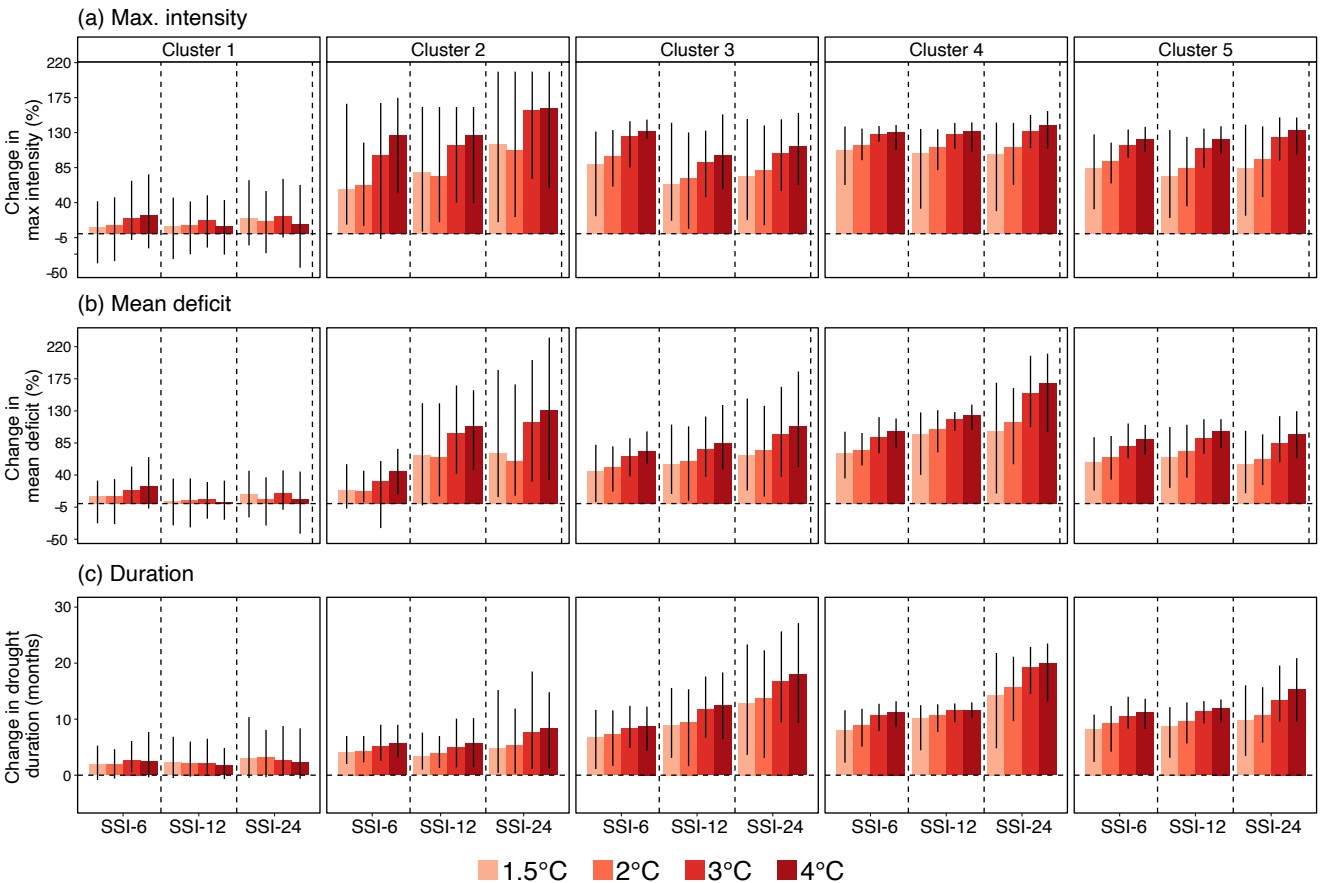

**Figure 12: Mean change across the 12 UKCP18 regional projections in a) max. drought intensity (%), b) mean drought deficit (%) and c) drought duration (months) for the 2010-2012 drought across four warming levels for each cluster and SSI accumulation period. Error bar indicates spread across the 12 regional projections.**

### 3.6 Comparison between storylines

To place the storylines in historical context, drought characteristics from the storylines are compared with two past droughts. We select the benchmark 1975-76 drought and the more protracted 1989-93 drought as comparators. Both droughts rank among the most severe since the 1970s (Marsh et al. 2007). Based on the characterization of severe droughts in the same set of catchments by Barker et al. (2019), the 1975-76 drought was the most severe in terms of maximum intensity and mean deficit across northeast Scotland and southern England (corresponding to Clusters 2 and 5), while the 1989-93 drought was

most severe for catchments in eastern England (corresponding to Cluster 4). Four storylines are selected to compare with past droughts – 1) "Driest preconditions", 2) "Dry year before", 3) "Dry year after" and 4) 2°C warming.

Figure 13 shows percentage change in max. intensity and mean deficit of the four storylines relative to the same characteristics calculated for the two past droughts. First, for the 1975-76 drought, drought conditions calculated using SSI-6 are in general

less severe across all four selected storylines. Cluster 1 is the exception where drought conditions match the 1975-76 drought for the "Dry year before" and "Driest precondition" storylines. When considering drought conditions at longer time scales using SSI-24, drought conditions of the four selected storylines exceed that of the 1975-76 drought for Clusters 3-5. The 2°C warming storyline (and warming levels beyond that) result in the largest increase out of the four selected storylines. For Clusters 1 and 2, drought conditions calculated using SSI-12 and 24 are less severe than the 1975-76 drought and less severe

than SSI-6. The "Dry year before" storyline for Cluster 1 is the exception where drought conditions exceed that of the 1975-76 drought for SSI-24 even though catchments in this cluster are fast responding.

Second, for the 1989-93 drought, conditions across the four selected storylines are estimated to be more severe apart from Cluster 4. Catchments in Cluster 4 were the most affected during the observed 1989-93 drought and only storylines with the

more extreme changes to the baseline drought could have led to similar or worse conditions than observed (i.e. "Driest preconditions" and 2°C and beyond warming). Out of the four storylines, a 2°C warming is estimated to result in the largest deviation from the 1989-93 drought for Clusters 3-5. The 2°C warming storyline is less severe for Clusters 1 and 2 where, respectively, the "Dry year before" and the "Driest preconditions" instead result in greater deviations from the 1989-93 drought. For all four selected storylines, the magnitude of change relative to the 1989-93 drought increases with accumulation

period and is greatest for SSI-24 for Clusters 3 and 5, indicating the importance of catchment memory.

In summary, the four storylines are all capable of causing more severe drought conditions for all clusters compared with the two past droughts. Conditions across the storylines are estimated to match the 1975-76 drought with comparatively more severe conditions for southern catchments at longer accumulation periods. Conditions are estimated to exceed the 1989-93

drought for all clusters apart from Cluster 4 which were the most affected in the observed event. Drought conditions decrease (increase) in severity with longer SSI accumulation periods for Clusters 1-2 (Clusters 3-5).

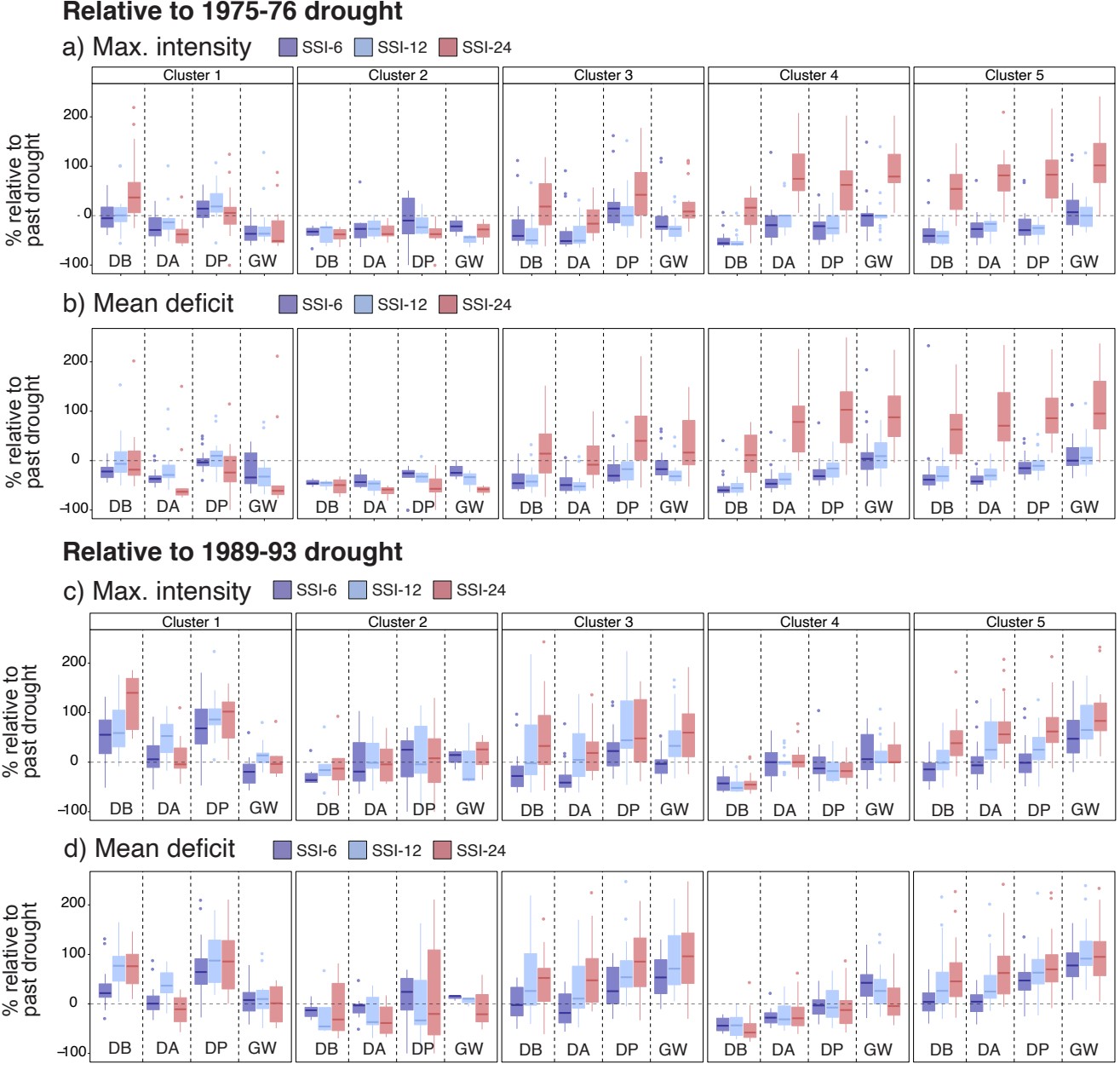

**Figure 13: Percentage difference in max. intensity (a and c) and mean deficit (b and d) calculated from SSI-6, 12 and 24 of selected storylines relative to the 1976-76 drought (top) and the 1989-93 drought (bottom). Drought characteristics of the past droughts are calculated from simulated river flow using the same input data for the baseline simulation as detailed in the methods section.**

## 4 Discussion

### 4.1 Hydrological drought risk

Drought characteristics of the 2010-12 drought support the northwest/southeast gradient for drought susceptibility identified

for multi-year droughts in Folland et al. (2015) and Barker et al. (2016). The five clusters correspond well with clusters identified in Barker et al. (2016), with upland catchments that were less permeable and southeastern catchments with higher storage, although the current study distinguishes an additional cluster distinguishing between catchments in southeastern and central England. Barker et al. (2016) also demonstrated that these clusters showed significant spatial variation in meteorological and hydrological drought characteristics for severe droughts since 1891.


The response to drier preconditions shows that the precondition persistence time vary across the clusters and can be differentiated by spatial variation in hydrogeology. The relationship with hydrogeology was also found in German and Swiss catchments by Stoelze et al. (2014) and Staudinger et al. (2015), where catchments at lower elevations which are generally flatter with the presence of porous aquifers were also found to have long response times after subjected to perturbations to

their initial conditions. The spatial variation in precondition persistence time (and hence catchment properties) confirms the importance of preconditions in determining the eventual timing and severity of the 2010-12 drought, with variation to the eventual drought characteristics between fast-responding northern and slow-responding southern catchments. Laaha et al. (2017) similarly concluded that preconditions of the preceding seasons in the 2003 and 2015 summer droughts played a crucial role in controlling the temporal and spatial dynamics, and that eventual event characteristics were modulated by both

preconditions and catchment properties. As the precondition persistence time only characterizes the influence of perturbations in the event's initial conditions, it is not indicative of full recovery from drought to non-drought conditions. Full drought recovery is a complex longer-term process that would require additional analyses of drought termination metrics such as that proposed in Parry et al. (2016). Results of the storylines of seasonal contributions, in particular, highlight the often-neglected role of autumn conditions in the development of multi-year droughts. Autumn conditions over the 2010-12 drought were

significant in controlling the timing of drought inception and termination, and considerably worsened drought conditions when coupled with consecutive years of dry winters.

Table 4 summarizes hydrological drought response for the storylines of temporal sequence and climate change. Placing the various storylines in context with a relatively short-term severe drought (1975-76) and a protracted multi-year drought (1989-

93) suggests that some or all of the selected storylines are estimated to result in are more severe drought conditions than the two past droughts. Conditions for all storylines could have exceeded that of both the 1975-76 and 1989-93 droughts even in catchments that were most severely affected in the observed droughts, particularly when considering long SSI accumulation periods. Comparison with the 1975-76 drought is consistent with findings in Burke et al. (2010) which placed future ensemble

projections in the context of the 1975-76 drought and concluded that the likelihood of future droughts with similar
characteristics to the 1975-76 drought can reach once every 10 years depending on the ensemble member considered.

**Table 4: Summary of drought response for fast and slow-responding catchments in the storylines of temporal sequence and climate change**

| Cluster | Location | Response | Hydrological drought response |
|---|---|---|---|
| 1 and 2 | E and W Scotland | Fast | - **Temporal sequence:** "Dry year before" highlights risk of intense drought in immediate response under progressively drier preconditions.<br>- **Climate change:** Drought projected to worsen with temperature rise. Change in intensity and deficit more pronounced for western Scotland. Conditions projected to be less severe at high warming levels due to wetter winters. |
| 3 | Midlands and SW England | Fast | - **Temporal sequence:** 2010-12 drought could have been more intense given an additional year with a third dry winter. "Dry year before" has a greater effect on drought characteristics although the "Dry year after" results in a greater increase in duration.<br>- **Climate change:** drought conditions projected to worsen with temperature rise, with particularly large increase in drought duration for the longer accumulation periods. |
| 4 and 5 | SE and Central England | Slow | - **Temporal sequence:** Conditions which were already the most affected in the 2010-12 drought would have been significantly worse without the dramatic termination in 2012. Observed preconditions were already dry that the repetition of a dry year prior to the event would have made little difference.<br>- **Climate change:** drought conditions projected to worsen with temperature rise with max. intensity and mean deficit both exceeding that of the 1975-76 drought beyond 2°C warming. |

It is interesting to consider the difference between UKCP18 and other projections such as the previous UKCP09 or CMIP5. Compared to UKCP09 and CMIP5 GCMs, UKCP18 projects a slightly larger reduction/smaller increase in precipitation during summer and autumn and greater warming during summer under the RCP8.5 scenario (Lowe et al. 2018). Precipitation is also projected to increase by a smaller magnitude in the winter compared to UKCP09. Recent analysis shows that the UKCP18 projections are better able to represent the observed spatial patterns of UK heatwaves than the CMIP5 models (Kennedy-Asser
et al. 2021). The regional projections used in the study are thus representative of worst-case scenarios that track the warmer end of the full range of outcomes for the RCP8.5 scenario in CMIP5. The storylines of seasonal contribution point towards an important role of autumn in the propagation of multi-year droughts. Compared to other projections, the smaller increase in autumn precipitation projected by UKCP18 may point to increased impacts of multi-year droughts developing as a result of autumn rainfall deficit, and drier conditions entering winter which take longer to recover and are thus more susceptible to the
development of multi-year droughts during dry winters.

## 4.2 Value of the storyline approach

The storyline approach represents a new research avenue to understand the impacts of unrealized droughts. Following the Water Act 2014, water companies are required to consider water supply reliability under plausible unobserved worst-case droughts (Environment Agency 2015a). One method is to use hydrological models to reconstruct historic river flows. Barker et al. (2019) used river flow reconstructions to identify the spatial coverage and hydrological characteristics of key pre-1961 droughts on a catchment-scale. A main drawback relates to hydrological model uncertainty and non-stationarity when faced with changes in climate and land use (Spraggs et al. 2015; Barker et al. 2019). An alternative method is to resample the observed record (e.g. Environment Agency 2015b) or generate synthetic meteorological sequences using stochastic weather generators in a response surface framework describing drought response from meteorological sequences that resemble incremental changes in certain statistical characteristics (e.g. Environment Agency 2013). However, challenges remain to verify the plausibility of synthetically generated droughts as they do not stem from actual drought events. Additionally, weather generators have predominantly been used as tools to statistically downscale coarse GCM projections for use in catchment hydrological models. Consequently, their use is associated with challenges such as uncertainty related to multi-site generation, the choice of statistical model and selection of evaluation/verification methods (Maraun et al. 2010).

Storylines represent an alternative way to consider specific stakeholder concerns on how catchments may respond in a given situation. Stoelze et al. (2020) and Hellwig et al. (2021) recently advocated for a catchment-scale recharge stress test framework, similar to the storylines of precondition severity in this study, to complement traditional climate change projections. Comparing the 12-month precipitation deficit of the storylines of precondition severity in this study with previous studies shows that they are comparable to the range considered in the H++ climate change scenarios for low rainfall and droughts (Wade et al. 2015). Additionally, the resulting 12-month rainfall deficit from each return period is also comparable to the rainfall deficit increments of the drought vulnerability framework. The drought vulnerability framework forms part of the guidance for water resources planning where statistically plausible droughts are produced using stochastic methods on a response surface for incremental variation in long term average rainfall (%) and drought duration (months) (Environment Agency 2020). When running water resources models to test management measures against long droughts by stacking multiple observed/reconstructed long duration droughts, Watts et al. (2012) emphasized that basing their analyses on actual events helped increase realism amongst decision-makers compared to stochastic or weather generator approaches. The storyline approach demonstrated here also builds on recent proposals to increase focus on event-based case study analyses that can better consider type II errors and combine multiple lines of evidence in the construction of plausible counterfactuals to inform risk management (Lloyd and Shepherd 2020; Sillmann et al. 2021). The in-depth comparative analysis by Laaha et al. (2017) of the 2003 and 2015 summer droughts demonstrated the potential for new insights to inform water management based on event-based studies.

Results from the various storylines in this study also confirm the potential of this approach to better consider and quantify worst case scenarios. For example, the "three dry winter" storylines in this study are able to consider the hypothetical, but plausible situation which could have seen dry conditions persist. This could have led to a higher maximum drought intensity and a lengthening of the drought, leading to drought characteristics approaching conditions as observed in past severe droughts. Motivated by similar aims as the "three dry winters" storylines in this study, water companies have previously considered the hypothetical situation of a third dry winter following the 2004-06 drought (Environment Agency 2011). The creation of event storylines by perturbing the observed drought sequence can be helpful when planning for droughts of high return periods (e.g. 1 in 500-years) for which estimates are highly uncertain with no observed historical precedent.

## 4.3 Limitations and future work

An event storyline approach could be used to assess counterfactuals of further imagined drought events. Additional insights may be obtained from contrasting drought events where minimum river flow occurs in different seasons (e.g. summer vs winter) or compound events associated with hydrological droughts (e.g. heatwave droughts). Storylines in this study are based on resampling and perturbing the meteorological time series of the 2010-12 drought. The delta change method used to place the 2010-12 drought in a warmer world retains the observed temporal variability of the observed drought. By not considering changes in the likelihood of such an event, it could under- or over-estimate drought impacts from climate change. However, in the absence of confident information on changes in likelihood of multi-year circulation anomalies, this is a logically sensible approach to take, grounded in Bayesian reasoning (Shepherd 2021). Alternative approaches can be used to consider natural variability and changes in wet and dry sequences under climate change.  For example, weather type analysis (e.g. Richardson et al. 2018) or meteorological analogues (e.g. Cattiaux et al. 2010) can provide a basis for imposing additional plausible changes to the event's drivers, if plausible storylines of such changes could be constructed. Storylines of extreme events under climate change can also be created in alternative ways. Recent studies have created event storylines using atmospheric nudging (e.g. Wehrli et al. 2020; van Garderen et al. 2021) or through searching for analogues resembling observed events (in both drivers and impacts) in large ensemble climate model data. Single model initial condition large ensembles (SMILEs) are well suited to construct event storylines as the larger sample size means a greater likelihood of finding analogue events with similar driving mechanisms to selected observed events (e.g. van der Wiel et al. 2021). Driving hydrological models with SMILEs can be a useful approach when considering changes in rare hydrological extremes not present in the observations (Brunner et al. 2021). Similarly, the Unprecedented Simulation of Extremes with ENsembles (UNSEEN) approach can be used to pool retrospective forecasts and large ensemble simulations to investigate storylines of unprecedented events in present and future climate (Thompson et al. 2017; Kelder et al. 2020).

Future work could also relate each storyline with management decisions through the use of water resource system models. This would require consideration of factors such as agricultural activities and water abstractions in relation to changes in reservoir yields. This was not included here as the majority of the selected catchments are not major catchments contributing

to public water supply. Additionally, as an extension to Smith et al. (2019) and Barker et al. (2019), this study employed the same hydrological model and parameter set to simulate hydrological response to each storyline. To account for hydrological model structural and more explicitly consider parameter uncertainty, the use of an ensemble of hydrological models and the entire suite of LHS500 parameter sets in Smith et al. (2019) would increase robustness of the results.

## 5 Conclusions

This study employs an event storyline approach to quantify "storylines" of how the 2010-12 UK drought could hypothetically have unfolded, or could unfold in the future. We extend previous work on historic droughts by applying the same set of hydrological models at catchments within the NRFA Low Flow Benchmark Network. Counterfactual storylines show the influence of preconditions and seasonal contributions on spatial and temporal drought development. Significant changes in drought characteristics from progressively drier preconditions highlight the importance of preconditions in controlling drought propagation and characteristics. The persistence of drier preconditions reflects both the spatial variation in drought characteristics and the role of catchment properties, especially hydrogeology, in drought propagation. Catchments across the UK remain vulnerable to a "third dry winter" situation as simulated by the "Dry year before" and "Dry year after" storylines. The "Dry year before" storyline shows that northern catchments are especially vulnerable to flash droughts in immediate response to dry winter conditions. The "Dry year after" storyline shows that drought conditions in southern catchments, which were already the most affected, could still have intensified significantly given continued dry conditions instead of the abrupt drought termination that actually occurred. UKCP18 climate change projections applied to the 2010-12 event point towards a decrease in mean streamflow across the majority of catchments at four global warming levels. Drought conditions are projected to intensify with temperature rise, with the exception of wetter winters for northern catchments mitigating drought conditions at high warming levels. Comparing drought characteristics of the multiple storylines shows how close the 2010-12 drought could have been to conditions observed in past severe droughts. Perturbations for all four sets of storylines could have resulted in drought conditions matching and exceeding that of both the benchmark 1975-76 and the 1989-93 droughts, particularly for catchments across southern England.

Although no probability is attached to each storyline, understanding outcomes from different storylines helps to navigate the cascade of uncertainty and reveals insights into hydrometeorological pathways of plausible drought events that may have significant implications for water resources. Return periods for each counterfactual storyline can also be estimated in combination with historical events (e.g. through extreme value analysis) to obtain further information on the severity of the counterfactual storylines.

## Author contributions

All authors were involved in the conceptualization of the study. WC conducted the formal analysis and prepared the original paper. TGS, KAS, GD and NWA supervised the study. All authors contributed to the writing and interpretation of the results.

## Competing interests

The authors declare no competing interests

## Data availability

Precipitation data (CEH-GEAR) is freely available on the Environmental Information Data Centre (Tanguy et al. 2019). Daily mean temperature (CEH-CHESS) is freely available on the Environmental Information Data Centre (Robinson et al. 2020). Daily observed river flow data is available from the National River Flow Archive (https://nrfa.ceh.ac.uk/). Calibration
parameters for the GR4J hydrological model at 303 UK catchments are available from the Environmental Information Data Centre (Smith et al. 2018). The input (precipitation and PET) and output (simulated river flows) data for each storyline for each catchment is available on the zenodo repository (https://doi.org/10.5281/zenodo.5180494).

## Acknowledgements

WC is funded by the Natural Environment Research Council (NERC) via the SCENARIO Doctoral Training Partnership (grant
NE/S007261/1). The authors would like to thank Cecilia Svensson (UKCEH) and Lucy Barker (UKCEH) for initial help in calculating the Standardized Streamflow Index (SSI). We also acknowledge the Editor and the two reviewers for their detailed comments on the analysis.

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
