# Peer review of "Storylines of UK drought based on the 2010-2012 event"

_Hydrology and Earth System Sciences, 2021_

## Referee Comment (RC2)

This paper assesses the impacts of different storylines of UK drought based on the 2010-2012 drought event. The results demonstrate the importance of meteorological preconditions, catchment characteristics controlling recovery time and the vulnerability of UK catchments to a 'three dry winter' scenario.

Overall I enjoyed reading the paper, it is nicely written and figures are well presented. There is some interesting analysis and conclusions that will be of great benefit to those working on drought in the UK and further afield. However, I do have some major comments for the authors to consider. In particular, some of the methods need clarification and better justification, and there needs to be more critical discussion and reflection on the use of storylines in drought analysis.

*Main Comments*

*Plausibility.* As noted in the introduction, 'Storylines are defined as physically self-consistent unfoldings of past events and the **plausible evolution** of these events in a future climate (Shepherd et al. 2018).'. I would like to challenge the authors and encourage more critical discussion in the manuscript on how 'plausible' the storyline scenarios are. You have implemented a number of different storylines but there is very little consideration of the plausibility of these storylines in terms of the atmospheric conditions that are needed to create them. Where is the evidence that you are implementing 'plausible' changes to this event that link to physical climate processes? What is the evidence that these are really '**physical** climate storylines'? You note that the 12month precipitation-deficits from the storylines are in line with other climate scenarios but many of your scenarios are based around precipitation deficits that span more than one year (i.e. up to three dry winters). The manuscript needs more critical discussion of the plausibility of the storylines and a fuller consideration of their limitations.

*Delta change approach.* Aligned with the comment above is the use of the delta change approach to represent changes in climate. There are a whole host of problems with delta change approaches (see Fowler et al, 2007 https://doi.org/10.1002/joc.1556) and again, in terms of plausibility, I think it is difficult to argue that applying mean monthly factors to a past drought event gives you a realistic picture of the 'hydrological impacts of climate change'. Again, there is no critical discussion of this in the paper.

*Estimating return periods.* In Section 2.2.1 you use annual average three month rainfall from 1965 – 2015 to estimate 10, 20, 50 and 100-year return periods. Firstly it is not clear what the source of this rainfall data is (I assume CEH-GEAR as this is referenced below?). Secondly, if it is CEH-GEAR (or Had-UK) then the rainfall data are available for much longer time periods (1890-2017). So why choose a shorter time period which could make your estimates less robust, particularly when you are trying to estimate a 1 in 100 year return period of rainfall?

*Catchment recovery time.* I don't really understand why you choose the baseline simulation as your threshold for the catchment recovery time. This isn't necessarily an indication of the catchment having 'recovered' – the baseline simulation may still be very low flows. Is the time calculated from the very beginning of the simulation? This metric needs to be better clarified and justified.

*Model Performance metrics.* Better justification for this choice of metrics is needed – what do they represent and why are they appropriate for this analysis? Should NSE (a metric focused on high flows) really be given equal weighting? Some maps of model performance (where dots are coloured by their best NSE/logNSE value for example) would be useful so we can see the spatial differences in model performance. I would expect more detailed analysis of how the model performs for the 2010-2012 event given the focus of the paper.

*Data Availability.* The data availability section needs to cover **all** the data used and produced in the paper. Will you be making the storyline input data available (i.e. the modified rainfall and temperature timeseries) for others to use? Will you be making the outputs available? This is important for reproducibility, transparency etc.

*Technical / Minor Comments*

L14. 'highly conditioned by its meteorological preconditions'. Not entirely sure what you mean here, can you clarify?

L55. You might also consider citing Dobson et al (https://doi.org/10.1029/2020WR027187) which considers the future spatial dynamics of droughts and water scarcity across England and Wales.

L116. It would be useful to add a map of the catchments (with the catchment boundaries) into the supplementary information. This would help highlight their size and spatial coverage across GB.

L150. 'The temporal variability of the reduced preconditions precipitation'. This doesn't make sense to me and should be reworded.

Figure 9 – how much variation is there in the percentage/absolute changes between the different clusters? i.e. are the projected changes in rainfall very different for cluster 1 compared to cluster 5? Might be worth adding these plots to the supplementary information for context as most of the subsequent analysis is focused on the changes for each cluster.

Figure 12 is quite blurry – can you increase the resolution?

---

## Author Comment (AC1)

**Author response**

General comments
The submitted manuscript addresses a very relevant topic for water risk management, (i.e. low likelihood/high impact events) and does so using storylines, a novel approach that allows the investigation of plausible but unrealized high impact events. The selected storylines are based on the 2010-2012 UK drought event and explore imposed changes to 1) Precondition severity, 2) Temporal drought sequence, and 3) Climate change. The implications of such changes are assessed by quantifying changes to streamflow maximum intensity, mean deficit, and duration. The results do not only facilitate the realization that it could have been worse/it possibly will be worse but also sheds light on physical catchment properties that play a key role in the propagation of a multi-year drought event. In general, the manuscript is well written and structured and the results are relevant to a broad community interested in novel approaches that tackle environmental risk management and future climate change impacts. I have few minor concerns that I share in what follows:

***RESPONSE: We thank the reviewer for the positive feedback on our manuscript. We are grateful for the comments and suggestions on how our manuscript can be improved. We respond to each comment given in the text below (in bold and italics text).***

Specific comments
I understand plausibility to be a key property of the designed storylines. The first storyline proposes varying 3- and 6- months prior precipitation conditions to the 2010-2012 drought event independently of other climatic variables used in the model simulation. Such manipulations do not consider correlation structures in the data. I find that not completely justified and slightly weakening the plausibility assumption. For example, the potential presence of autocorrelation among successive monthly precipitation values or the correlation between precipitation and temperature are not considered. The authors can potentially mention these concerns in their discussion to further strengthen the plausibility argument.

***RESPONSE: We agree that further information is needed to discuss the implications of the precondition storylines on the correlation between potential evapotranspiration (PET) and precipitation. Figure R1 (below) shows monthly precipitation and PET from 1965-2015. Apart from a slight negative correlation between precipitation and PET in spring and summer, there appears to be no clear correlation between the variables in the remaining months from 1965-2015 data. Figure R2 shows the equivalent values after precipitation 3- (i.e. OND 2009) and 6-months (i.e. JASOND 2009) before the 2010-12 drought reduced to match mean OND or JASOND precipitation at four return periods. The reduction to precipitation prior to the drought does not appear to be outliers compared to the observed relationship between precipitation and PET from 1965-2015. We also emphasize that the creation of event-based storylines in other locations should consider potential correlation between the different variables if a strong correlation is found. We will amend the text to reflect this. Figure R1 will be added in the supplementary materials and Figure R2 in the main text of the revised manuscript.***

*Regarding the reviewer's other concern, monthly precipitation values show very low autocorrelation among successive monthly precipitation values in all years (1965-2015). The average autocorrelation for successive monthly precipitation values across all years is -0.046 with values falling within the 95% confidence interval, indicating no statistical significance. As there is low autocorrelation, we believe the reductions to precipitation applied in the storylines of precondition severity are justified.*

[Figure]

**Figure R1 Observed relationship between PET and precipitation for each month for the period 1965-2015 averaged across the 100 UK catchments selected with the correlation coefficient value shown for each month.**

[Figure]

*Figure R2 October to December monthly precipitation and PET (1965-2015) (top) and July to December monthly precipitation and PET (1965-2015) (bottom) The black circle indicates observed value in 2009 while the colored circles indicate the value after the precipitation 3- (top) and 6-months (bottom) prior to the 2010-12 drought is reduced at four return periods.*

I see that some consideration is given in the paragraph starting at Line 516, nevertheless, I find that rather short and in itself not fully convincing. If I understand correctly, the authors address plausibility for the precondition storylines by comparing the resultant 12-month precipitation deficits to outputs of high-end climate change scenarios. They argue that the preconditioning storylines are plausible as these are contained within the range of outputs from high-end climate change scenarios. Nevertheless, I expected that plausibility concerning these particular storylines should address whether such conditions are possible in the current climate.

*RESPONSE: The Environment Agency vulnerability framework and the high-end H++ climate change scenarios were intended as a point of comparison when*

*discussing the implications of the storylines of precondition severity instead of a justification of their plausibility. We will amend and move this text to Section 4.2 in our discussion of the value of the storyline approach to highlight these storylines as alternatives to existing projections. As discussed in the previous response, we have expanded our justification of the plausibility of the precondition storylines.*

The authors state that they apply the delta approach in its standard form (line 189) where historical variability is retained. This formulation confuses me a bit as I am not sure what a non-standard form for the delta approach is.

*RESPONSE: The standard form of the delta/change factor approach, as applied in this study, retains historical variability with monthly change factors. There have been different modifications or variations to the delta approach proposed in the literature. They mostly consist of ways to calculate percentile- or quantile-based change factors for relative changes in wet and dry days and rainfall intensity (e.g. Anandhi et al. 2011; Willems and Vrac 2011; Ntegeka et al. 2014). Anandhi et al. (2011) also reviews and presents a classification of different variants of the change factor method. Although there are several modifications, the standard delta method as used in this paper remains the most widely used. We will clarify this by referencing these studies in the revised manuscript. In response to the next comment, we will also expand on the limitations of this method and discuss other alternative methods.*

Can the authors expand on this in their discussion to address limitations associated with the method they chose and possibly elaborate on other potential methods that can be used to answer questions such as: How would that particular event look like in a warmer world? (e.g. Wehrli et al. 2020).

*RESPONSE: We will expand on limitations relating to the use of the delta method. The obvious limitation is that the delta method omits the influence of changes in wet/dry sequences. While this is a limitation to the storylines of climate change, the other storylines created in this study considers alternative changes to the wet/dry sequence of the observed drought.*

*Thanks for pointing us to Wehrli et al. (2020). We will add the citation as suggested as well as van Garderen et al. (2021) which is also an example of the use of atmospheric nudging as an alternative method to construct event-based storylines of extreme events in a warmer climate. Alternative approaches to investigate extreme events in a warmer world would be to search for analogues or events similar to the 2010-12 drought (for example, analysis of weather types or circulation patterns – e.g. Cattiaux et al. 2010, or through the use of large ensemble climate model data – e.g. van der Wiel et al. 2020). We will amend the text in the revised manuscript in the limitations and future work section to highlight these alternative approaches.*

*In response to a previous point, we will also add additional justification of the change factor method in the methods section. This particularly relates to challenges associated with the statistical bias correction and downscaling techniques, all of which assume that the biases corrected for and the bias*

*adjustment technique remain valid for future time periods. We believe the delta change method is suitable for this study as retaining the observed temporal sequence of the 2010-12 drought increases realism and enables quick comparison with the other storylines which were also created based on altering the observed time series of the 2010-12 drought.*

It is clear to me why storylines are relevant as complementary information to already existing approaches that rely on GCM projections to quantify the hydrological impacts of climate change. I also do understand how these two approaches are very much different in scope. Nevertheless, the authors use the terms "scenario-driven approach" as a particular feature of GCM driven assessments in an attempt to contrast their approach and I find that slightly misleading. Storylines are still very much scenarios to my understanding, event-based in that case, and with a focus on plausibility rather than probability. I don't see why they wouldn't qualify as scenario-driven. The author themselves state that (i.e. line 143): "storylines follow similar methodologies employed in previous studies to create scenarios". I, therefore, recommend revisiting specifically this phrasing to reduce confusion and facilitate the understanding of what is meant by storylines.

*RESPONSE: The term "storylines" has been used in various ways in different disciplines. We will clarify our definition of "storylines" used in this paper. "Storylines" as used here refer to event-based "physical climate storylines" with a focus on plausible changes to causal elements of the 2010-12 drought. This differs from traditional GCM-driven climate change impact assessments which are constrained by GCM projections following different emissions or socio-economic scenarios (e.g. SSPs, RCPs etc.) when used as inputs to impact models. Their results are therefore constrained by the choice of the different scenarios and GCMs. "Physical climate storylines" can be created independently from these to represent situations or conditions that could lead to significant impacts and complement results from GCM-driven approaches. We will clarify the differences between existing approaches and our creation of "physical climate storylines" throughout the paper. We have also explained what was meant by our use of "scenario-driven" more explicitly.*

*We agree that line 143 may be confusing to readers. We consider the similar methodologies in the previous studies as cited to also be storylines that could be used to complement climate change projections and stress test hydrological systems. We will remove any mention of scenarios in this case.*

Another point related to terminology: Can the authors explain their use of the term "counterfactual" when discussing future impacts of climate change. As the climate change storyline refers to a hypothetical event in the future, I find it a bit unclear why that would qualify as a counterfactual.

*RESPONSE: We agree with the reviewer on the potentially confusing use of the term "counterfactual" when discussing climate change. We will remove any use of the term when discussing the use of the UKCP18 climate projections.*

**Technical comments**

1.  I am slightly confused by this sentence: Line 373, "The drought is estimated to worsen for the "Dry year before" storyline for all clusters except for mean drought deficit for Cluster 4 for SSI-6". I believe something along the lines of: "The drought defined by SSI-6 is estimated to worsen for the "Dry year before" storyline for all clusters except for mean drought deficit for Cluster 4 " is a bit more clear.
    *RESPONSE: Thanks for the suggestion. We will modify the text as suggested in the revised paper.*

2.  Line 378: is -> are
    *RESPONSE: We will change this in the revised paper.*

3.  Line 381-382: I believe something in the punctuation of the phrase is incorrect. Please check that.
    *RESPONSE: We will modify this in the revised paper.*

*References*

*Anandhi, A., Frei, A., Pierson, D. C., Schneiderman, E. M., Zion, M. S., Lounsbury, D., and Matonse, A. H.: Examination of change factor methodologies for climate change impact assessment, Water Resources Research., 47, https://doi.org/10.1029/2010WR009104, 2011.*

*Ntegeka, V., Baguis, P., Roulin, E., and Willems, P.: Developing tailored climate change scenarios for hydrological impact assessments, Journal of Hydrology., 508, 307–321, https://doi.org/10.1016/j.jhydrol.2013.11.001, 2014.*

*van der Wiel, K., Selten, F. M., Bintanja, R., Blackport, R. and Screen, J. A.: Ensemble climate-impact modelling: extreme impacts from moderate meteorological conditions, Environ. Res. Lett., 15(3), 034050, doi:10.1088/1748-9326/ab7668, 2020.*

*van Garderen, L., Feser, F., and Shepherd, T. G.: A methodology for attributing the role of climate change in extreme events: a global spectrally nudged storyline, Natural Hazards and Earth System Sciences., 21, 171–186, https://doi.org/10.5194/nhess-21-171-2021, 2021.*

*Wehrli, K., Hauser, M., and Seneviratne, S. I.: Storylines of the 2018 Northern Hemisphere heatwave at pre-industrial and higher global warming levels, Earth Syst. Dynam., 11, 855–873, https://doi.org/10.5194/esd-11-855-2020, 2020.*

*Willems, P. and Vrac, M.: Statistical precipitation downscaling for small-scale hydrological impact investigations of climate change, Journal of Hydrology.,402, 193–205 https://doi.org/10.1016/j.jhydrol.2011.02.030, 2011.*

---

## Author Comment (AC2)

**Referee 2**
**Author Response**

1. General comments
   This paper assesses the impacts of different storylines of UK drought based on the 2010-2012 drought event. The results demonstrate the importance of meteorological preconditions, catchment characteristics controlling recovery time and the vulnerability of UK catchments to a 'three dry winter' scenario. Overall I enjoyed reading the paper, it is nicely written and figures are well presented. There is some interesting analysis and conclusions that will be of great benefit to those working on drought in the UK and further afield. However, I do have some major comments for the authors to consider. In particular, some of the methods need clarification and better justification, and there needs to be more critical discussion and reflection on the use of storylines in drought analysis

*RESPONSE: We thank Dr. Coxon for the positive feedback and suggestions on how our manuscript can be improved. We are grateful that the reviewer agrees that our results have benefit to those working on droughts. We respond to each comment given in the text below (in bold and italics).*

2. Main comments
   Plausibility. As noted in the introduction, 'Storylines are defined as physically self-consistent unfoldings of past events and the plausible evolution of these events in a future climate (Shepherd et al. 2018).'. I would like to challenge the authors and encourage more critical discussion in the manuscript on how 'plausible' the storyline scenarios are. You have implemented a number of different storylines but there is very little consideration of the plausibility of these storylines in terms of the atmospheric conditions that are needed to create them. Where is the evidence that you are implementing 'plausible' changes to this event that link to physical climate processes? What is the evidence that these are really 'physical climate storylines'? You note that the 12month precipitation-deficits from the storylines are in line with other climate scenarios but many of your scenarios are based around precipitation deficits that span more than one year (i.e. up to three dry winters). The manuscript needs more critical discussion of the plausibility of the storylines and a fuller consideration of their limitations.

*RESPONSE: A similar point was also raised by reviewer 1 who was concerned about the plausibility of altering observed precipitation independently of temperature in the storylines of precondition severity. Fig. R1 shows the observed relationship between monthly precipitation and PET from 1965-2015 which shows no clear correlation apart from a slight negative correlation in spring and summer. This shows that our precipitation perturbations are plausible and do not violate any correlation structures between precipitation and temperature. This is further shown by Fig. R2 (addressing temporal correlations) which shows the equivalent values after precipitation 3- (i.e. OND 2009) and 6-months (i.e. JASOND 2009) before the 2010-12 drought reduced to match OND and JASOND precipitation at four return periods, and which are seen to fall within the historical relationships. Both figures will be included in*

*the revised manuscript. We will also emphasize in the revised manuscript that the creation of event-based storylines in other locations should consider potential correlation between the different variables if a strong correlation is found. The Environment Agency vulnerability framework and the high-end H++ climate change scenarios were intended as a point of comparison when discussing the implications of the storylines of precondition severity instead of a justification of their plausibility. We will make this clearer in the revised manuscript.*

[Figure]

***Figure R1** Observed relationship between PET and precipitation for each month for the period 1965-2015 averaged across the 100 UK catchments selected with the correlation coefficient value shown for each month.*

[Figure]

*Figure R2 October to December monthly precipitation and PET (1965-2015) (top) and July to December monthly precipitation and PET (1965-2015) (bottom) The black circle indicates observed value in 2009 while the colored circles indicate the value after the precipitation 3- (top) and 6-months (bottom) prior to the 2010-12 drought is reduced at four return periods.*

*With regard to precipitation perturbations over a longer time period, a large number of previous studies have investigated the occurrence and likelihood of sequences of dry winters and their implications for UK water resources. Past multi-year droughts have been shown to include at least one dry winter (Environment Agency 2009; Watts et al. 2012; Folland et al. 2015). The effects of successive dry winters have been shown through multiple reconstructions of precipitation, river flows and groundwater levels across the UK (e.g. Spraggs et al. 2015; Barker et al. 2019; Watts et al. 2012; Bloomfield et al. 2019). In river flow reconstructions, a "third dry winter" scenario was shown to cause significant reduction in storage at key reservoirs in East Anglia during 1943-46 (Spraggs et al. 2015). Similarly, precipitation reconstructions showed that even longer dry spells are plausible with "the long drought" between 1890-1910 characterized*

*by three or more successive dry winters punctuated by wet interludes (Marsh et al. 2007). Quantifying transition probabilities of consecutive dry seasons, Wilby et al. (2015) found that the longest spell of consecutive dry winter or summer half-years spanned 4 years (including 4 dry winters) in the 1870s in the England and Wales Precipitation (EWP) time series. The same study also found the longest observed sequence of consecutive river flow deficit reached 5.5 years during the 1988-93 drought in southern England.*

*Motivated by similar aims as the "three dry winters" storyline in this study, water companies have previously considered the hypothetical situation of a third dry winter following the 2004-06 drought which was characterized by two consecutive dry winters (Environment Agency 2009). Similarly, there was widespread concern and expectation in early 2012 that dry conditions would persist based on the prevailing atmospheric conditions at the time (Bell et al. 2013; Spraggs et al. 2015). The repetition of a dry year to represent continued dry conditions is therefore a reasonable and plausible case to investigate given concerns at the time. Figure R3 shows more clearly the differences in atmospheric circulation between the repeated year and the year replaced. Average geopotential height at 500hPa (Z500) anomalies from ERA5 for 2010 (repeated year) compared with 2012 (replaced year) show that summer and winter 2010 was characterized by high pressure over parts of the UK throughout the year. Conversely, 2012 was characterized by low pressure over the UK and high rainfall totals across 2012 which terminated the drought. We will expand and cite the studies above to better justify the plausibility of the three dry winter storyline in the revised manuscript.*

[Figure]

*Figure R3 Summer and winter geopotential height (m) at 500 hPa (Z500) anomalies relative to 1978-2015 from ERA5 for 2010 (left) and 2012 (right). The average monthly rainfall totals (mm) are shown in the corner for the respective year.*

3. Delta change approach. Aligned with the comment above is the use of the delta change approach to represent changes in climate. There are a whole host of problems with delta change approaches (see Fowler et al, 2007 https://doi.org/10.1002/joc.1556) and again, in terms of plausibility, I think it is difficult to argue that applying mean monthly factors to a past drought event gives you a realistic picture of the 'hydrological impacts of climate change'. Again, there is no critical discussion of this in the paper.

*RESPONSE There are a whole host of problems associated with bias correction and downscaling (Maraun et al. 2017), with the realism of climate model simulations (especially for persistent circulation extremes), and with knowing how atmospheric circulation will respond to climate change (Shepherd 2014). There is no easy answer here, and nobody can claim to predict the future under such conditions. The storyline approach sacrifices generality for physical plausibility. In particular, our aim here is to place the 2010-12 drought in a future climate instead of generalizing the hydrological impacts of climate change across dissimilar drought events. We believe the delta change method is suitable for this as retaining the baseline temporal sequence of the 2010-12 drought increases realism and enables quick comparison with the other storylines which were also created based on altering the observed time series of the 2010-12 drought.*

*Although the delta method omits the influence of changes in wet/dry sequences in the storylines of climate change, the other storylines created in this study consider changes to the wet/dry sequence of the observed drought. Despite the limitations of the delta change approach, it remains widely used in hydrological climate change impact assessments globally. It also remains the most widely used method for UK catchments and has been used consistently since the 1990s to reach important conclusions on the potential impacts of climate change on UK water resources. We will expand on this justification of the delta change method in the methods section in the revised manuscript. There are alternative emerging methods available to place historical events under future warming. This includes searching for analogue events (e.g. Cattiaux et al. 2010), the use of large ensemble climate model data (e.g. van der Wiel et al. 2020) or atmospheric nudging of climate models (e.g. van Garderen et al. 2021). We will expand on these alternative approaches in the revised manuscript.*

4. Estimating return periods. In Section 2.2.1 you use annual average three month rainfall from 1965 – 2015 to estimate 10, 20, 50 and 100-year return periods. Firstly it is not clear what the source of this rainfall data is (I assume CEH-GEAR as this is referenced below?). Secondly, if it is CEH-GEAR (or Had-UK) then the rainfall data are available for much longer time periods (1890- 2017). So why choose a shorter time period which could make your estimates less robust, particularly when you are trying to estimate a 1 in 100 year return period of rainfall?

*RESPONSE: Apologies for the confusion, the precipitation data we used was CEH-GEAR. We will amend the typo in the data availability section in the revised manuscript. We chose the time period of 1965-2015 as the baseline period as we*

*did not have temperature (and PET) data for the longer time period for hydrological modelling. However, we agree that estimates could be more robust with the full dataset. As suggested by the reviewer's comments, we can revise our estimates of rainfall return periods using the full CEH-GEAR dataset and amend Figures 5 and 6 accordingly.*

*It should be noted that the aims for the storylines of precondition severity are not to improve the estimates of rainfall totals at a particular return period but rather to investigate sensitivity of different catchments to various magnitudes of rainfall perturbations. We believe this aim was satisfied with the estimates of return periods using the shorter period (1965-2015) and do not anticipate changes to the overall conclusions with return periods calculated from the full dataset.*

5. Catchment recovery time. I don't really understand why you choose the baseline simulation as your threshold for the catchment recovery time. This isn't necessarily an indication of the catchment having 'recovered' – the baseline simulation may still be very low flows. Is the time calculated from the very beginning of the simulation? This metric needs to be better clarified and justified.

*RESPONSE: The aim of quantifying the "catchment recovery time" was to investigate how long the influence of different precondition perturbations persists for each catchment and how that might relate to physical catchment characteristics. The metric was calculated from the start of the perturbation until the influence of the perturbation is no longer detected (<1% compared with baseline). Based on the reviewer's comments, we realize the use of the term might be confusing. We will define this better in the revised manuscript to clarify that the catchment recovery time as calculated in this study is not indicative of how long the catchment took to recover from drought conditions but is instead indicative of how long the influence of precondition perturbations lasts for each catchment. A similar experimental set-up was also proposed in Staudinger and Seibert (2014) which calculated a catchment "relaxation" and "persistence" time from perturbations to initial conditions. This is also consistent with Stoelzle et al. (2020) which proposed the evaluation of drought stress tests using what they termed "catchment recovery duration" which is defined in the same way as in this study.*

6. Model Performance metrics. Better justification for this choice of metrics is needed – what do they represent and why are they appropriate for this analysis? Should NSE (a metric focused on high flows) really be given equal weighting? Some maps of model performance (where dots are coloured by their best NSE/logNSE value for example) would be useful so we can see the spatial differences in model performance. I would expect more detailed analysis of how the model performs for the 2010-2012 event given the focus of the paper.

*RESPONSE: We agree that there should be more clarification and discussion of model performance during past droughts and the 2010-12 drought. The model parameters were taken from Smith et al. (2019). In that study, the authors used a Latin hypercube sampling calibration approach across the selected metrics and demonstrated the model performance of the top 500 parameter sets. The*

*authors also showed that periods of drought identified from simulated river flow match observed occurrence of past droughts well. This study takes the top 500 parameter sets from Smith et al. (2019) and re-ranks them based on performance during dry years. We will clarify this in the methods section to justify how the parameter sets and calibration strategy are appropriate for drought analysis. As suggested, we will include maps of NSE and logNSE in the supplementary material. We also propose to include Figure R4 which shows simulated river flow during the 2010-12 drought for nine example catchments spread across the UK, showing the ability of the model to reproduce low river flows across the catchments during this period.*

[Figure]

*Figure R4 Daily observed (black) and simulated (red) river flow across nine example catchments from the top parameter set in re-ranked parameter ensemble from Smith et al. (2019). The y axis is presented in log scale.*

7. Data Availability. The data availability section needs to cover all the data used and produced in the paper. Will you be making the storyline input data available (i.e. the modified rainfall and temperature timeseries) for others to use? Will you be making the outputs available? This is important for reproducibility, transparency etc.

*RESPONSE: As the aim of our study was not to create a new dataset, we did not describe the input and output data in the data availability section. However, if the editor and reviewer believe that the input and output data could potentially be of interest to the community, we can make the modified rainfall and temperature time series and the simulation outputs for each storyline available in the interest of transparency (possibly via zenodo or similar repository?).*

**Technical comments**

1. L14. 'highly conditioned by its meteorological preconditions'. Not entirely sure what you mean here, can you clarify?
   ***RESPONSE: We will rephrase. What we meant was that the spatial and temporal characteristics of the 2010-12 drought were highly influenced by the meteorological conditions 3- and 6-months prior to drought inception.***

2. L55. You might also consider citing Dobson et al (https://doi.org/10.1029/2020WR027187) which considers the future spatial dynamics of droughts and water scarcity across England and Wales.
   ***RESPONSE: Thanks for pointing us to Dobson et al. (2020). We will cite this in the revised manuscript as suggested.***

3. L116. It would be useful to add a map of the catchments (with the catchment boundaries) into the supplementary information. This would help highlight their size and spatial coverage across GB.
   ***RESPONSE: Thanks for the useful suggestion. We will add this to the supplementary information.***

4. L150. 'The temporal variability of the reduced preconditions precipitation'. This doesn't make sense to me and should be reworded.
   ***RESPONSE: We will rephrase this.***

5. Figure 9 – how much variation is there in the percentage/absolute changes between the different clusters? i.e. are the projected changes in rainfall very different for cluster 1 compared to cluster 5? Might be worth adding these plots to the supplementary information for context as most of the subsequent analysis is focused on the changes for each cluster.
   ***RESPONSE: We will add a supplementary figure on projected change in rainfall across the different clusters as suggested. We anticipate this will broadly reflect differences between the clusters in changes in drought characteristics in the storylines of climate change section (i.e. Fig.11).***

6. Figure 12 is quite blurry – can you increase the resolution?
   ***RESPONSE: We will modify this.***

***References***

***Barker, L. J., Hannaford, J., Chiverton, A. and Svensson, C.: From meteorological to hydrological drought using standardised indicators, Hydrology and Earth System Sciences, 20(6), 2483–2505, doi:https://doi.org/10.5194/hess-20-2483-2016, 2016.***
***Bloomfield, J. P., Marchant, B. P., and McKenzie, A. A.: Changes in groundwater drought associated with anthropogenic warming, 23, 1393–1408, https://doi.org/10.5194/hess-23-1393-2019, 2019.***
***Environment Agency: Impact of long droughts on water resources: https://www.gov.uk/government/publications/impacts-of-long-droughts-on-water-resources (last access: 21 June 2021), 2009.***
***Folland, C. K., Hannaford, J., Bloomfield, J. P., Kendon, M., Svensson, C., Marchant, B. P., Prior, J., and Wallace, E.: Multi-annual droughts in the English Lowlands: a***

review of their characteristics and climate drivers in the winter half-year, 19, 2353–2375, https://doi.org/10.5194/hess-19-2353-2015, 2015.

Maraun, D., Shepherd, T. G., Widmann, M., Zappa, G., Walton, D., Gutiérrez, J. M., Hagemann, S., Richter, I., Soares, P. M. M., Hall, A., and Mearns, L. O.: Towards process-informed bias correction of climate change simulations, 7, 764–773, https://doi.org/10.1038/nclimate3418, 2017.

Marsh, T., Cole, G. and Wilby, R.: Major droughts in England and Wales, 1800–2006, Weather, 62(4), 87–93, doi:10.1002/wea.67, 2007.

Shepherd, T. G.: Atmospheric circulation as a source of uncertainty in climate change projections, 7, 703–708, https://doi.org/10.1038/ngeo2253, 2014.

Smith, K. A., Barker, L. J., Tanguy, M., Parry, S., Harrigan, S., Legg, T. P., Prudhomme, C., and Hannaford, J.: A multi-objective ensemble approach to hydrological modelling in the UK: an application to historic drought reconstruction, 23, 3247–3268, https://doi.org/10.5194/hess-23-3247-2019, 2019.

Spraggs, G., Peaver, L., Jones, P., and Ede, P.: Re-construction of historic drought in the Anglian Region (UK) over the period 1798–2010 and the implications for water resources and drought management, Journal of Hydrology, 526, 231–252, https://doi.org/10.1016/j.jhydrol.2015.01.015, 2015.

Staudinger, M. and Seibert, J.: Predictability of low flow – An assessment with simulation experiments, Journal of Hydrology, 519, 1383–1393, https://doi.org/10.1016/j.jhydrol.2014.08.061, 2014.

Stoelzle, M., Staudinger, M., Stahl, K., and Weiler, M.: Stress testing as complement to climate scenarios: recharge scenarios to quantify streamflow drought sensitivity, in: Proceedings of the International Association of Hydrological Sciences, Hydrological processes and water security in a changing world - Hydrological Processes and Water Security in a Changing World, Beijing, China, 6–9 November 2018, 43–50, https://doi.org/10.5194/piahs-383-43-2020, 2020.

van der Wiel, K., Selten, F. M., Bintanja, R., Blackport, R. and Screen, J. A.: Ensemble climate-impact modelling: extreme impacts from moderate meteorological conditions, Environ. Res. Lett., 15(3), 034050, doi:10.1088/1748-9326/ab7668, 2020.

van Garderen, L., Feser, F., and Shepherd, T. G.: A methodology for attributing the role of climate change in extreme events: a global spectrally nudged storyline, Natural Hazards and Earth System Sciences., 21, 171–186, https://doi.org/10.5194/nhess-21-171-2021, 2021.

Watts, G., Christierson, B. von, Hannaford, J. and Lonsdale, K.: Testing the resilience of water supply systems to long droughts, Journal of Hydrology, 414–415, 255–267, https://doi.org/10.1016/j.jhydrol.2011.10.038, 2012.

Wilby, R. L., Prudhomme, C., Parry, S., and Muchan, K. G. L.: Persistence of Hydrometeorological Droughts in the United Kingdom: A Regional Analysis of Multi-Season Rainfall and River Flow Anomalies, J. of Extr. Even., 02, 1550006, https://doi.org/10.1142/S2345737615500062, 2015.

---

## Referee Report (RR1)

Thanks for the authors for expanding on plausibility assumptions and clarifying key terms in their revisions. I think the paper can now be accepted subject to a few technical corrections.

Line 39: Missing a full stop after deficits.

Line 99: Reference missing for Doblas-Reyes et al. 2021 in the reference list, please double check references.

Line 374: I believe you meant to say "alone" rather than "along" ?

---

## Referee Report (RR2)

Many thanks to the author for considering my comments and for the effort in the responses. I really appreciate the additional analyses and figures in the papers and supplementary information.

I have two minor comments for the authors that relate to my original review and need further discussion in the paper:

1. Model performance metrics

Thanks for the additional detail in the supplementary material and text in the methods. However, the justification for the choice of metrics still needs improving. In the paper, it is stated that *'The metrics selected are unweighted as high flows (NSE), timing of flows (logNSE), flow variability (MAPE) and overall water balance (PBIAS) should be considered equally important for river flows during the driest years.'* However, there is still no discussion (as far as I can tell) of **why** they should be considered important and **why** they should be considered equally important. Why are high flows important to consider when analysing future storylines of drought?

2. Delta change approach

I still feel that there is a lack of critical discussion related to the delta change approach in the limitations section. While I appreciate the additional text on alternative methods, there is only one sentence (as far as I can tell – apologies if I have missed this) on the limitations of a delta change approach: *'By not considering changes in the likelihood of such an event, it could under- or over-estimate drought impacts from climate change.'* There needs to be more critical discussion of this approach and its ability to capture how droughts might unfold in the future.

---

## Author Response (AR2)

**Response to all reviewers Storylines of UK drought based on the 2010-12 event**

**Response to Editor**

We thank Prof. Jim Freer for the comments and suggestions on how to improve the manuscript. We address each point raised by the Editor and reviewers below (in red). The revised manuscript with tracked changes is attached below the responses.

1) In response to your request for clarification on the publication of the story lines data then I recommend you read our publication policy on data used in papers where it is quite clear we expect this to be part and parcel of published papers now for improvements to transparency and scrutiny. I think you will find this is what most international journals are expecting. Please see https://www.hydrology-and-earth-system-sciences.net/policies/data\_policy.html

**RESPONSE: Thank you. We have made the input and output data publicly available via the zenodo repository (available at: https://doi.org/10.5281/zenodo.5180494)**

2) I am not yet sold on your arguments for only using delta change as the bias correction technique. In papers that I have been associated with in Flooding we specifically didn't not use delta change because we were looking at more extremes and remained unconvinced that basic delta change was valuable to reflect fully those extreme behaviour biases. I would suspect that the biases in the extremes associated with drought would be so similarly impacted and thus some hard analyses needs to be implemented in your paper to show that delta change would not unduly influence the storyline results if one scrutinized the extremes a little more. As you know I wrote to you about this in the beginning that this needs to be challenged and it has been brought up in the review process. So I don't think you can justify this for extreme event behaviour by saying it's commonly used in the UK without some further hard evidence of it's impact (or not). I'm super happy to be proved that this is not important to your methods. We all recognize there is no perfect answer to bias corrections and the deficits of downscaling and RCM precipitation quantities but that doesn't mean that should be the core justification for not exploring these issues for your own best scientific scrutiny given the core aims of your paper.

RESPONSE: The aim of the climate change component of our analysis (which we must emphasize, is only one component) is to place the 2010-12 drought in a warmer world rather than to generalize over the hydrological impacts of climate change. We sacrifice generality by focusing solely on the observed event but what we gain from this is interpretability and physical realism (further discussed in the next paragraph). We quote Reviewer 1 in this respect, who agrees that our analysis "sheds light on physical catchment properties that play a key role in the propagation of multi-year drought event". The underlying principle here is Box's famous "All models are wrong, some are useful". The delta change method allows us to do this by perturbing the observed drought sequence directly. Ensuring future plausibility is inherently difficult. While more complex statistical bias correction techniques such as quantile mapping can used to assess general changes in the impacts of climate change on hydrological variables (although we note that there are many known limitations to more complex bias adjustments: e.g. Ehret et al. 2012; Maraun et al. 2017), it is much more challenging to use bias corrected climate model data to search for similar analogues to observed events. This is particularly the case for droughts (and perhaps different from flooding applications, which tend to be shorter term weather rather than longer term climatic events). This is because of concerns and uncertainty over the realism of climate model simulations for persistent circulation extremes, and with how atmospheric circulation will respond to climate change (Shepherd 2014). Previous studies have shown that climate models tend to underestimate drought persistence (particularly important for multi-year droughts like the 2010-12 drought) and where multi-year droughts are simulated, the driving mechanisms in the climate model can vary significantly between individual drought events (Ault et al. 2014; Moon et al. 2018) hence making it difficult to validate in relation to the observed 2010-12 event. If we have perfect models and large enough ensembles, storyline approaches won't be needed. Single model initial condition large ensembles may be used to search for analogue events (e.g. van der Wiel 2021) but this is subject to on-going work and is out of the scope of this study.

We therefore believe using the change factor method for the climate change component of the paper is justified here as it has the additional advantage of being easily interpretable and comparable with the other storylines analysed in this paper, which were also created by altering the observed drought sequence. As all the event-based storylines created in this study were based on the observed drought sequence, we believe this actually increases realism compared to searching for dissimilar events in bias-corrected climate model data from models that cannot reproduce the persistent circulation anomalies that lead to the 2010-12 drought. In practice, altering the observed drought sequence in this way is valuable for water resources planning as it allows for the exploration of droughts at high return periods for which there is no historical precedent and could complement approaches following existing Environment Agency guidelines. We have added our justification in the methods section to explain how we believe the change factor method is suitable for this study (lines 231-250 in the revised manuscript). We previously raised the point that the delta method has been used widely and consistently in the UK for further context to support our choice, rather than as the core justification.

3) I also agree with comments made about the calibration of the model metrics not being very well justified or seemingly related to the core issue that the models produce historical drought behaviour well. The reasons and justification for these equally weighted metrics (which seem ad hoc at this moment) must be improved and that may need some additional analyses. I'm sure you have your reasons and analysed this in more detail than the paper currently shows, so we need that more intelligent calibration approach better identified.

RESPONSE: The Editor and Reviewer 2 are concerned about how the four selected metrics were selected and whether the fact that they are equally weighted would affect the choice of the parameters to simulate droughts. The ability of the top 500 parameter sets (LHS500) to reproduce periods of historic drought in both timing and magnitude has already been demonstrated in Smith et al. (2019). As the original LHS500 was ranked based on model performance over a long baseline period, a differential split-sample test is conducted by reranking LHS500 based on performance during the driest years using four of the six metrics in Smith et al. (2019). As we're calculating the metrics for river flow during the driest years, we don't believe any of the four metrics can be considered more important than the others as high flows (NSE), timing of flows (logNSE), variability (MAPE) and overall water balance (PBIAS) during dry years are all equally important. Model performance for the top parameter set in the original LHS500 rank and the Dry rank are comparable but, in some catchments, the Dry rank show better performance during the driest years. The top ranked parameter set in the original LHS500 ranking remains unchanged in the Dry rank for 17 out of the 100 catchments. For the majority of catchments (54 out of 100), the top parameter set in the new Dry rank is within the top 10 of the original LHS500 rankings. For the remaining catchments, the top parameter set in the new Dry rank are all found in the top 100 of the original LHS500 rankings. We have

added an additional figure (Supplementary Figure S4c) comparing the top parameter set in the Dry rank and its corresponding position in the original LHS500 ranking.

More generally, however, the details of the model parameter calibration process are not critical to the findings of this paper. We are seeking to apply a model which produces plausible hydrological simulations corresponding to the range in catchment and climate conditions across the UK. The key point is whether the model is informative and helps us evaluate the storyline concept – and we believe it does. As Reviewer 2 recommended, we have also provided two additional figures in the supplementary materials. Fig S4 shows high NSE and logNSE values for catchments across the UK using the top parameter set of the Dry rank. Fig S5 shows simulated river flows across 2010 and 2012 and clearly shows that the model is able to reproduce low river flows and drought conditions.

4) Finally the reviewers have asked for improvements to the justification of the recovery time metrics. This was another issue I need in my editorial review of your initial paper. I am still confused as to how a simple on any one day threshold metric has value to look at drought recovery which is a longer term process. Is there really no more intelligent way to approach this that deals with that longer term process? I'd like to see more critical discussion of that please.

RESPONSE: In our initial response to the Reviewer's comments on this, we have clarified that the catchment recovery time as calculated in the study is not indicative of the time taken for catchments to fully recover from drought to non-drought conditions. Instead, it is meant as an indicator for how long the influence of the precondition perturbations were felt for each catchment. This is consistent with similar indices proposed to investigate the impacts of changes in initial conditions and we have added those references in the revised text. To avoid further misunderstanding, catchment recovery time is renamed precondition persistence time which more accurately reflects what is shown in the results. The Editor is correct that other metrics would be needed if the aims are to consider full recovery from drought to non-drought conditions. As this was not the aim of the storylines of precondition severity, we point the reader to Parry et al. (2016) for calculating drought termination metrics to achieve this.

**Response to Reviewer 1**

**General comments**

The submitted manuscript addresses a very relevant topic for water risk management, (i.e. low likelihood/high impact events) and does so using storylines, a novel approach that allows the investigation of plausible but unrealized high impact events. The selected storylines are based on the 2010-2012 UK drought event and explore imposed changes to 1) Precondition severity, 2) Temporal drought sequence, and 3) Climate change. The implications of such changes are assessed by quantifying changes to streamflow maximum intensity, mean deficit, and duration. The results do not only facilitate the realization that it could have been worse/it possibly will be worse but also sheds light on physical catchment properties that play a key role in the propagation of a multi-year drought event. In general, the manuscript is well written and structured and the results are relevant to a broad community interested in novel approaches that tackle environmental risk management and future climate change impacts. I have few minor concerns that I share in what follows:

RESPONSE: We thank the reviewer for the positive feedback on our manuscript. We are grateful for the comments and suggestions on how our manuscript can be improved. We respond to each comment below (in red).

I understand plausibility to be a key property of the designed storylines. The first storyline proposes varying 3- and 6- months prior precipitation conditions to the 2010-2012 drought event independently of other climatic variables used in the model simulation. Such manipulations do not consider correlation structures in the data. I find that not completely justified and slightly weakening the plausibility assumption. For example, the potential presence of autocorrelation among successive monthly precipitation values or the correlation between precipitation and temperature are not considered. The authors can potentially mention these concerns in their discussion to further strengthen the plausibility argument.

RESPONSE: We agree that further information is needed to discuss the implications of the precondition storylines on the correlation between potential evapotranspiration (PET) and precipitation. We have added two new figures to address this comment (Figure 1 in the revised manuscript and Figure S2 in the supplementary materials). Figure S2 in the supplementary materials shows monthly precipitation and PET from 1965-2015. Apart from a slight negative correlation between precipitation and PET in spring and summer, there appears to be no clear correlation between the variables in the remaining months from 1965-2015 data. Figure 1 in the revised manuscript shows the equivalent values after precipitation 3- (i.e. OND 2009) and 6-months (i.e. JASOND 2009) before the 2010-12 drought was increased/reduced to match mean OND or JASOND precipitation at four return periods. The changes in precipitation prior to the drought does not appear to be outliers compared to the observed relationship between precipitation and PET from 1965-2015. We also emphasize that the creation of event storylines in other locations outside the UK should consider potential correlation between the different variables if a strong correlation is found.

Autocorrelation among successive monthly rainfall values is mostly not statistically significant (within the 95th confidence interval) apart from the short-term and decays rapidly after the first 1-2 months. For some stations, there are statistically significant but low autocorrelation values highlighting rainfall seasonality. Low monthly autocorrelation for rainfall is also seen in previous studies when considering the performance of stochastic weather generators (e.g. Kilsby et al. 2007; Serinaldi and Kilsby 2012; Chun et al. 2013). Given the low autocorrelation found in the observed data, the 3- and 6-months perturbations in the storyline of precondition severity do not violate existing autocorrelation structures and are valid and plausible. We have amended the text to reflect this.

I see that some consideration is given in the paragraph starting at Line 516, nevertheless, I find that rather short and in itself not fully convincing. If I understand correctly, the authors address plausibility for the precondition storylines by comparing the resultant 12-month precipitation deficits to outputs of high-end climate change scenarios. They argue that the preconditioning storylines are plausible as these are contained within the range of outputs from high-end climate change scenarios. Nevertheless, I expected that plausibility concerning these particular storylines should address whether such conditions are possible in the current climate.

RESPONSE: The Environment Agency vulnerability framework and the high-end H++ climate change scenarios were intended as a point of comparison when discussing the implications of the storylines of precondition severity, rather than as a justification of their plausibility. We

have amended and moved this text to Section 4.2 in our discussion of the value of the storyline approach to highlight these storylines as alternatives to existing projections. As discussed in the previous response, we have expanded our justification of the plausibility of the precondition storylines.

The authors state that they apply the delta approach in its standard form (line 189) where historical variability is retained. This formulation confuses me a bit as I am not sure what a non-standard form for the delta approach is.

RESPONSE: The standard form of the change factor approach, as applied in this study, retains historical variability with monthly change factors. There have been different modifications or variations to the delta approach proposed in the literature. They mostly consist of ways to calculate percentile- or quantile-based change factors for relative changes in wet and dry days and rainfall intensity (e.g. Anandhi et al. 2011; Willems and Vrac 2011; Ntegeka et al. 2014). Anandhi et al. (2011) also reviews and presents a classification of different variants of the change factor method. Although there are several modifications, the standard delta method as used in this paper remains the most widely used. We have clarified this by referencing these studies in the revised manuscript.

Can the authors expand on this in their discussion to address limitations associated with the method they chose and possibly elaborate on other potential methods that can be used to answer questions such as: How would that particular event look like in a warmer world? (e.g. Wehrli et al. 2020).

RESPONSE: We have expanded on the limitations relating to the use of the delta method. We have added reference to Wehrli et al. (2020) and van Garderen et al. (2021) as examples of spectral nudging. We have also expanded on other methods to construct event storylines under climate change. Alternative approaches to investigate extreme events in a warmer world would be to search for analogues or events similar to the 2010-12 drought (for example, analysis of weather types or circulation patterns – e.g. Cattiaux et al. 2010, or through the use of large ensemble climate model data – e.g. van der Wiel et al. 2021).

The validity of the change factor method was also raised by Reviewer 2 and the Editor. We have expanded on the reasons for choosing the delta method in the Methods section in the revised manuscript (see earlier response to Editor; lines 231-250 in revised manuscript).

It is clear to me why storylines are relevant as complementary information to already existing approaches that rely on GCM projections to quantify the hydrological impacts of climate change. I also do understand how these two approaches are very much different in scope. Nevertheless, the authors use the terms "scenario-driven approach" as a particular feature of GCM driven assessments in an attempt to contrast their approach and I find that slightly misleading. Storylines are still very much scenarios to my understanding, event-based in that case, and with a focus on plausibility rather than probability. I don't see why they wouldn't qualify as scenario-driven. The author themselves state that (i.e. line 143): "storylines follow similar methodologies employed in previous studies to create scenarios". I, therefore, recommend revisiting specifically this phrasing to reduce confusion and facilitate the understanding of what is meant by storylines.

RESPONSE: It is true that in lay usage, "storyline" and "scenario" are somewhat interchangeable terms. However, in climate change science, the word "scenario" is firmly

established as corresponding to a specified socio-economic pathway associated with a particular climate forcing, and to use it in any other sense would cause unnecessary confusion. Shepherd et al. (2018) discussed this issue as part of their rationale for the use of the term "storyline" in the context of physical climate, and their definition of physical climate storyline has now been adopted by the IPCC Glossary. Here we specifically use the concept of a physical climate "event storyline", as illustrated in Box 10.2, Figure 1 of the AR6 WG1 report (available from <a href="https://www.ipcc.ch/report/ar6/wg1/#FullReport">https://www.ipcc.ch/report/ar6/wg1/#FullReport</a>). We have now clarified our language and made reference to the IPCC usage of these words. We would argue that from this perspective, it is not misleading to distinguish between scenario-driven and storyline approaches. Physical climate storylines can be created independently of scenario-driven GCM ensemble projections to represent situations or conditions that could lead to significant impacts, and can complement results from scenario-driven GCM ensemble projections. To avoid misunderstanding, we have checked all our uses of "scenario" and "storyline" and provided clarifications in each case.

We agree that line 143 may be confusing to readers. We consider the similar methodologies in the previous studies as cited to also be storylines that could be used to complement climate change projections and stress test hydrological systems. We have removed any mention of "scenario" in this case.

Another point related to terminology: Can the authors explain their use of the term "counterfactual" when discussing future impacts of climate change. As the climate change storyline refers to a hypothetical event in the future, I find it a bit unclear why that would qualify as a counterfactual.

RESPONSE: We agree with the reviewer that the use of the term "counterfactual" when discussing the storylines of climate change is potentially confusing. All use of the term when discussing the UKCP18 climate projections have been removed.

Technical comments

I am slightly confused by this sentence: Line 373, "The drought is estimated to worsen for the "Dry year before" storyline for all clusters except for mean drought deficit for Cluster 4 for SSI-6". I believe something along the lines of: "The drought defined by SSI-6 is estimated to worsen for the "Dry year before" storyline for all clusters except for mean drought deficit for Cluster 4 " is a bit more clear.

RESPONSE: We have rephrased as suggested.

Line 378: is -> are **RESPONSE: Done**.

Line 381-382: I believe something in the punctuation of the phrase is incorrect. Please check that.

RESPONSE: We have modified this.

**Response to Reviewer 2**

General comments

This paper assesses the impacts of different storylines of UK drought based on the 2010-2012 drought event. The results demonstrate the importance of meteorological preconditions,

catchment characteristics controlling recovery time and the vulnerability of UK catchments to a 'three dry winter' scenario. Overall I enjoyed reading the paper, it is nicely written and figures are well presented. There is some interesting analysis and conclusions that will be of great benefit to those working on drought in the UK and further afield. However, I do have some major comments for the authors to consider. In particular, some of the methods need clarification and better justification, and there needs to be more critical discussion and reflection on the use of storylines in drought analysis

RESPONSE: We thank Dr. Coxon for the positive feedback and suggestions on how our manuscript can be improved. We are grateful the reviewer agrees that our results have benefit to those working on droughts. We respond to each comment given in the text below (in red).

**Main comments**

Plausibility. As noted in the introduction, 'Storylines are defined as physically self-consistent unfoldings of past events and the plausible evolution of these events in a future climate (Shepherd et al. 2018).'. I would like to challenge the authors and encourage more critical discussion in the manuscript on how 'plausible' the storyline scenarios are. You have implemented a number of different storylines but there is very little consideration of the plausibility of these storylines in terms of the atmospheric conditions that are needed to create them. Where is the evidence that you are implementing 'plausible' changes to this event that link to physical climate processes? What is the evidence that these are really 'physical climate storylines'? You note that the 12month precipitation-deficits from the storylines are in line with other climate scenarios but many of your scenarios are based around precipitation deficits that span more than one year (i.e. up to three dry winters). The manuscript needs more critical discussion of the plausibility of the storylines and a fuller consideration of their limitations.

RESPONSE: A similar point was raised by Reviewer 1 who was concerned about the plausibility of altering observed precipitation independently of temperature in the storylines of precondition severity. In response to that comment, we have added Figure 1 in the revised manuscript and Figure S1 in the supplementary materials to illustrate that our perturbation of precipitation 3- and 6-months prior to the observed drought does not violate any correlation structures between precipitation and potential evapotranspiration (PET). We have also emphasized in the revised manuscript in the methods section that the creation of event-based storylines in other locations should consider potential correlation between different variables if a strong corelation is found. The discussion of the H++ high end climate change scenarios as intended as a point of comparison rather than as a justification of our storylines of precondition severity. We have moved this to the discussion.

With regard to precipitation perturbations over a longer time period with the "three dry winter" storylines, we consider this to be plausible as a large number of previous studies have investigated the occurrence and likelihood of sequences of dry winters and their implications for UK water resources. We have added additional background on the "three dry winters" situation in the methods section. In addition to this, we also added reference to a previous Environment Agency study which looked at a similar storyline with a third dry winter following the 2004-06 drought. There was widespread concern and expectation in early 2012 that dry conditions would persist based on the prevailing atmospheric circulation before abrupt record-breaking rainfall terminated the drought. An additional figure has been added to the supplementary material (Figure S2) which shows more clearly the differences in atmospheric circulation between the repeated year and the year replaced. Average Z500 anomalies for 2010 (repeated year) show were characterized by high pressure over parts of the UK throughout

summer and winter. In contrast, 2012 (replaced year) was characterized by low pressure over the UK and high rainfall totals throughout the year. The repetition of a dry year to represent continued dry conditions is therefore a reasonable and plausible case to investigate given concerns at the time. We have added this justification in the methods section. Greater consideration of atmospheric conditions will be the subject of future work.

Delta change approach. Aligned with the comment above is the use of the delta change approach to represent changes in climate. There are a whole host of problems with delta change approaches (see Fowler et al, 2007 https://doi.org/10.1002/joc.1556) and again, in terms of plausibility, I think it is difficult to argue that applying mean monthly factors to a past drought event gives you a realistic picture of the 'hydrological impacts of climate change'. Again, there is no critical discussion of this in the paper.

RESPONSE: This relates to the limitations of the delta change method also raised by the Editor and Reviewer 1. The reviewer is concerned about the plausibility of applying the delta change method. However, we would argue that it actually increases realism with the additional advantage of increasing familiarity to stakeholders by retaining the observed drought sequence. A key characteristic of event storylines like the ones created in this study is that they are familiar and link directly to experiences of stakeholders (as highlighted in Box 10.2 in the IPCC AR6 report). Please also see earlier response to Editor for an expanded justification on this.

We think that characterizing Fowler et al.'s paragraph on delta-change methods as identifying "a whole host of problems" is overstating what is said in that paper. A number of caveats are mentioned (not specifically about droughts), but they are only caveats, and every method has its caveats. These caveats are also all discussed in more detail now in the revised manuscript. The assumption that GCMs simulate relative changes better than absolute values is the cornerstone of the analysis presented in IPCC reports, so it is not a radical assumption. We have expanded on our discussion in the methods section to explain why we believe the change factor method is suitable for this study. We have also expanded on this and other alternative approaches in the limitations section in the Discussion.

Estimating return periods. In Section 2.2.1 you use annual average three month rainfall from 1965 – 2015 to estimate 10, 20, 50 and 100-year return periods. Firstly it is not clear what the source of this rainfall data is (I assume CEH-GEAR as this is referenced below?). Secondly, if it is CEH-GEAR (or Had-UK) then the rainfall data are available for much longer time periods (1890- 2017). So why choose a shorter time period which could make your estimates less robust, particularly when you are trying to estimate a 1 in 100 year return period of rainfall?

RESPONSE: The dataset used was CEH-GEAR. We have amended the typo in the data availability section. We agree that our estimates of precipitation return periods could be most robust with the full CEH-GEAR dataset from 1900. It should be noted that the aims for the storylines of precondition severity are not to improve the estimates of rainfall totals at a particular return period but rather to investigate sensitivity of different catchments to various magnitudes of rainfall perturbations. As suggested by the reviewer's comments, we have repeated the simulations for this section based on revised return periods calculated using the full dataset. This resulted in updates to Figures 5 and 6 in the original manuscript (Figures 6 and 7 in the revised manuscript) and corresponding changes to the discussion of the results.

Catchment recovery time. I don't really understand why you choose the baseline simulation as your threshold for the catchment recovery time. This isn't necessarily an indication of the

catchment having 'recovered' – the baseline simulation may still be very low flows. Is the time calculated from the very beginning of the simulation? This metric needs to be better clarified and justified.

RESPONSE: In response to comments from the reviewers and the Editor, we have clarified that the catchment recovery time as calculated in the study is not indicative of the time taken for catchments to recover from drought to non-drought conditions. Instead, it is meant as an indicator of how long the influence of the precondition perturbations were felt for each catchment. This is consistent with similar indices proposed to investigate the impacts of changes in initial conditions. To avoid further misunderstanding, catchment recovery time is renamed precondition persistence time, which more accurately reflects what is shown in the results. The metric is calculated from the start of the perturbation until the influence of the perturbation is no longer detected (<1% compared with baseline). We have clarified this in the revised manuscript with references to Stoelze et al. (2020) and Staudinger and Seibert (2014) which used similar metrics to understand the influence of changes to initial conditions. Although our aim was not to investigate the time taken for the catchment to fully recover from drought to non-drought conditions, we have added reference to Parry et al. (2016) which outlines a framework to calculate drought termination metrics.

Model Performance metrics. Better justification for this choice of metrics is needed – what do they represent and why are they appropriate for this analysis? Should NSE (a metric focused on high flows) really be given equal weighting? Some maps of model performance (where dots are coloured by their best NSE/logNSE value for example) would be useful so we can see the spatial differences in model performance. I would expect more detailed analysis of how the model performs for the 2010-2012 event given the focus of the paper.

RESPONSE: This point was also raised by the Editor. Please see prior response to Editor for expanded justification. As recommended, we have added supplementary Figures S4 and S5 NSE/logNSE values for each catchment for the top parameter set in the Dry rank and simulated river flow across the 2010-12 drought. Simulated river flow matches well with the observations in both timing and magnitude of low flows across 2010 and 2012. The NSE and logNSE values are high for the top-ranked parameter set in the Dry rank (>0.5) across most of the catchments. logNSE values are generally higher than NSE values. As the Dry rank is based on ranking parameter sets based on the driest years, NSE values (which as the reviewer noted is focused on high flows) are lower as a result. Catchments with relatively poorer performance are highlighted in the original text as fast-responding catchments with a "flashy" river regime in northern Scotland as identified in Smith et al. (2019).

Data Availability. The data availability section needs to cover all the data used and produced in the paper. Will you be making the storyline input data available (i.e. the modified rainfall and temperature timeseries) for others to use? Will you be making the outputs available? This is important for reproducibility, transparency etc.

RESPONSE: We have made the input and output data available via the zenodo repository (available at: https://doi.org/10.5281/zenodo.5180494).

Technical comments

L14. 'highly conditioned by its meteorological preconditions'. Not entirely sure what you mean here, can you clarify?

RESPONSE: We have rephrased. What we meant was that the spatial and temporal characteristics of the 2010-12 drought were highly influenced by the meteorological conditions 3- and 6-months prior to drought inception.

L55. You might also consider citing Dobson et al (https://doi.org/10.1029/2020WR027187) which considers the future spatial dynamics of droughts and water scarcity across England and Wales.

RESPONSE: Thanks for pointing us to Dobson et al. (2020). We have cited this in the revised manuscript as suggested.

L116. It would be useful to add a map of the catchments (with the catchment boundaries) into the supplementary information. This would help highlight their size and spatial coverage across GB.

RESPONSE: Thanks for the useful suggestion. We have added this as Figure S1 in the supplementary materials.

L150. 'The temporal variability of the reduced preconditions precipitation'. This doesn't make sense to me and should be reworded. RESPONSE: We have rephrased this.

Figure 9 – how much variation is there in the percentage/absolute changes between the different clusters? i.e. are the projected changes in rainfall very different for cluster 1 compared to cluster 5? Might be worth adding these plots to the supplementary information for context as most of the subsequent analysis is focused on the changes for each cluster.

RESPONSE: We have added a Figure S11 in the supplementary materials on projected change in mean annual precipitation across the different clusters as suggested. This broadly reflect differences between the clusters in changes in drought characteristics in the storylines of climate change section (i.e. Fig.12 in the revised manuscript).

Figure 12 is quite blurry – can you increase the resolution? **RESPONSE: Done.**

**Combined references**

- Anandhi, A., Frei, A., Pierson, D. C., Schneiderman, E. M., Zion, M. S., Lounsbury, D. and Matonse, A. H.: Examination of change factor methodologies for climate change impact assessment, Water Resources Research, 47(3), doi:10.1029/2010WR009104, 2011.
- Ault, T. R., Cole, J. E., Overpeck, J. T., Pederson, G. T., and Meko, D. M.: Assessing the Risk of Persistent Drought Using Climate Model Simulations and Paleoclimate Data, 27, 7529–7549, https://doi.org/10.1175/JCLI-D-12-00282.1, 2014.
- Cattiaux, J., Vautard, R., Cassou, C., Yiou, P., Masson-Delmotte, V. and Codron, F.: Winter 2010 in Europe: A cold extreme in a warming climate, Geophysical Research Letters, 37(20), doi:10.1029/2010GL044613, 2010.
- Ehret, U. et al. (2012). Should we apply bias correction to global and regional climate model data? *Hydrol. Earth Syst. Sci.*, 16: 3391–3404. doi:10.5194/hess-16-3391-2012.
- Maraun, D. et al. (2017) Towards process-informed bias correction of climate change simulations. *Nature Climate Change* 7, 764-773

- Ntegeka, V., Baguis, P., Roulin, E., and Willems, P.: Developing tailored climate change scenarios for hydrological impact assessments, 508, 307–321, https://doi.org/10.1016/j.jhydrol.2013.11.001, 2014.
- Parry, S., Wilby, R. L., Prudhomme, C., and Wood, P. J.: A systematic assessment of drought termination in the United Kingdom, 20, 4265–4281, https://doi.org/10.5194/hess-20-4265-2016, 2016.
- Shepherd, T. G.: Atmospheric circulation as a source of uncertainty in climate change projections, Nature Geoscience, 7(10), 703–708, doi:10.1038/ngeo2253, 2014.
- Staudinger, M. and Seibert, J.: Predictability of low flow An assessment with simulation experiments, Journal of Hydrology, 519, 1383–1393, https://doi.org/10.1016/j.jhydrol.2014.08.061, 2014.
- Smith, K. A., Barker, L. J., Tanguy, M., Parry, S., Harrigan, S., Legg, T. P., Prudhomme, C. and Hannaford, J.: A multi-objective ensemble approach to hydrological modelling in the UK: an application to historic drought reconstruction, Hydrology and Earth System Sciences, 23(8), 3247–3268, doi:https://doi.org/10.5194/hess-23-3247-2019, 2019.
- Stoelzle, M., Stahl, K., Morhard, A. and Weiler, M.: Streamflow sensitivity to drought scenarios in catchments with different geology, Geophysical Research Letters, 41(17), 6174–6183, doi:10.1002/2014GL061344, 2014.
- Moon, H., Gudmundsson, L., and Seneviratne, S. I.: Drought Persistence Errors in Global Climate Models, 123, 3483–3496, https://doi.org/10.1002/2017JD027577, 2018.
- van der Wiel, K., Lenderink, G., and de Vries, H.: Physical storylines of future European drought events like 2018 based on ensemble climate modelling, 33, 100350, https://doi.org/10.1016/j.wace.2021.100350, 2021.
- Willems, P. and Vrac, M.: Statistical precipitation downscaling for small-scale hydrological impact investigations of climate change, 402, 193–205, https://doi.org/10.1016/j.jhydrol.2011.02.030, 2011.

**Relevant changes to the manuscript:**

- Further justification of the storyline approach in relation to existing approaches with reference to "physical climate event storylines" recently adopted in the IPCC AR6 Glossary
- Simulations for the storylines of precondition severity were re-ran based on estimations of rainfall return periods using the entire CEH-GEAR dataset (1900-2015) including changes to Figures 6 and 7 in the revised manuscript
- Further consideration and justification for the plausibility of the storylines of precondition severity (including Figure 1 in revised manuscript) and the "three dry winter" storylines
- Expanded discussion and justification of the delta change method used to construct storylines of climate change
- Expanded discussion on alternative approaches to construct event storylines
- Minor changes to other sections and figures

---

## Author Response (AR3)

**Response to all reviewers**
**Storylines of UK drought based on the 2010-12 event**

**Reviewer 1**

Thanks for the authors for expanding on plausibility assumptions and clarifying key terms in their revisions. I think the paper can now be accepted subject to a few technical corrections.

Line 39: Missing a full stop after deficits.
Line 99: Reference missing for Doblas-Reyes et al. 2021 in the reference list, please double check references.
Line 374: I believe you meant to say "alone" rather than "along" ?

Response: We thank the reviewer for reviewing our manuscript again and for the positive comments and suggestions. The corrections have all been made.

**Reviewer 2**

Many thanks to the author for considering my comments and for the effort in the responses. I really appreciate the additional analyses and figures in the papers and supplementary information. I have two minor comments for the authors that relate to my original review and need further discussion in the paper:

Response: We thank the reviewer for reviewing our manuscript again and for the positive comments and suggestions. We respond to the two minor comments below (in red).

1. Model performance metrics
Thanks for the additional detail in the supplementary material and text in the methods. However, the justification for the choice of metrics still needs improving. In the paper, it is stated that 'The metrics selected are unweighted as high flows (NSE), timing of flows (logNSE), flow variability (MAPE) and overall water balance (PBIAS) should be considered equally important for river flows during the driest years.' However, there is still no discussion (as far as I can tell) of why they should be considered important and why they should be considered equally important. Why are high flows important to consider when analysing future storylines of drought?

Response: We believe that high flows, timing of low flows, flow variability and overall water balance for river flows during dry years are equally important and the parameter set used to evaluate the storylines should represent not just low river flows but the full range of flow response during dry years (including the possibility of wetter interludes during dry years). This is especially relevant for the 2010-12 drought as there were wetter periods during the drought period and the drought itself was abruptly terminated by unexpected wet conditions with high river flows (as discussed in section 3.1). The full range of flow response is also important to understand the antecedent conditions of drought events. This is important given that antecedent conditions to the 2010-12 drought is varied in the storylines of precondition severity. Using the top parameter set in the new Dry rank, Supplementary Fig.S4 shows clearly that the parameter sets are capable of simulating both the high and low flows during the 2010-12 drought period across the example catchments spread across the UK. The revised justification in the text now reads as follow (lines 239-248):

*"As the LHS500 ranking was based on model performance over a long baseline period, we conduct a differential split-sample experiment to re-rank LHS500. For each catchment, the 10 driest years were selected based on mean annual precipitation (1965-2015). Model performance for each of the driest years was calculated using daily observed and simulated river flow for four of the metrics in Smith et al. (2019): NSE, logNSE, MAPE and PBIAS. The metrics selected are unweighted as high flows (NSE), timing of low flows (logNSE), flow variability (MAPE) and overall water balance (PBIAS) should be considered equally important for river flows during the driest years. This is to ensure that the full range of flow response during dry years is considered, including the potential for wetter interludes during dry years (such as that seen during the 2010-12 drought). It is also important to consider high flows during the antecedent conditions of drought events. In the context of the storyline approach, this is especially important given that antecedent conditions are varied in the storylines of precondition severity."*

2. Delta change approach

I still feel that there is a lack of critical discussion related to the delta change approach in the limitations section. While I appreciate the additional text on alternative methods, there is only one sentence (as far as I can tell – apologies if I have missed this) on the limitations of a delta change approach: 'By not considering changes in the likelihood of such an event, it could underor over-estimate drought impacts from climate change.' There needs to be more critical discussion of this approach and its ability to capture how droughts might unfold in the future.

Response: The fact that the delta method retains the observed meteorological time series is often cited as the main limitation of the approach (as reflected in the limitations section and also stated in the section 2.2.3). We have restructured and expanded the limitations section to reflect on the implications of this in understanding how future droughts may unfold (lines 584-600). The revised text now reads as follow:

*"Storylines in this study are based on resampling and perturbing the meteorological time series of the 2010-12 drought. The main limitation of the delta change method used to place the 2010-12 drought in a warmer world is that it retains the observed temporal variability of the observed drought. This approach is advantageous given the specific focus on the 2010-12 drought, and it avoids having to deal with potential climate mode biases in the representation of the persistent circulation anomalies that lead to drought. However, the temporal variability and sequencing of weather events may change under climate change and future changes in variability differs between GCM/RCMs. The delta method applied to the 2010-12 drought therefore means that we do not consider ways in which a drought of different nature could unfold in the future and reach similar or worse impacts as the 2010-12 drought. Future droughts where minimum river flow occurs in different seasons (e.g. summer vs winter) or driven by compound conditions (e.g multivariate heatwave drought or preconditioned drought from combination of seasonal precipitation deficits) (Zscheischler et al. 2020) thus cannot be assessed using the delta change method alone. By not considering changes in the likelihood of such an event, the delta change method could therefore under- or over-estimate drought impacts from climate change. For example, Wilby and Harris (2006) has previously shown that the direct use of statistically downscaled climate model output can lead to a smaller reduction in low flows and a wider range of projected change compared to the delta change method although overall uncertainty is dominated by differences between GCMs. However, given GCM-related uncertainty and in the absence of confident information on changes in the likelihood of multi-year circulation anomalies, using the delta change method to place a*

*singular event under future warming is a logically sensible approach to take, grounded in Bayesian reasoning (Shepherd 2021)."*

**Reviewer 3**

The paper (from my point of view) is now way too long (more than 700lines and 13 Fig.). I recommend that a part of the analyses is removed to streamline the paper (e.g. focus on one instead of two storylines?). However, the topic and the idea of the paper is really important for the community and I lean towards a 2nd round of revisions and not towards a rejection. A lot of confusing terms have been removed in the first round of reviews, this is nice to see and there is some progress in the paper. Unfortunately, the paper is now somehow inflated and should be narrowed down again to one (or two) key messages.

Response: We thank the reviewer for reviewing our manuscript. We are grateful the reviewer agrees that the paper is useful for the community and that there is progress after revisions made after the first round of reviews. We have reviewed and have shortened the overall length of the paper. We have removed one figure from the main text (Fig 10, now Fig.S10) and the length of the paper is now close to the length of the original submitted manuscript and contains the same number of figures.

We do not think that any of the storylines should be removed as all the storylines considered explore different aspects of the 2010-12 drought. The storylines of seasonal contribution show the importance of autumn conditions in the 2010-12 drought and should be considered in assessment of future multi-year droughts. The storylines of precondition severity show the sensitivity of the drought to plausible drier preconditions that are linked to both the spatial dynamics of the drought and also catchment characteristics (e.g. hydrogeology). The "Dry year before" and "Dry year after" storylines show distinctive drought responses between the fast- and slow-responding catchments and represent plausible alternatives of the observed drought. The storylines of climate change using the UKCP18 projections place the drought at different global warming levels and showed worsening of conditions with temperature rise. We showed that all storylines are capable of matching or exceeding conditions observed in past droughts and are thus important elements to consider in the unfolding of future high-impact drought events.

Fig.1: This figure is added to the manuscript after first round of reviews. The explanation given (L190-204) is helpful to understand how the severity of preconditions is assessed. However, I am not totally convinced that PET (as response variable on the y-axis) is the best choice (variability is rather small), perhaps AET would be better (i.e. possibility to identify situations where the systems are water-limited).

Response: The figure was added after reviewers 1 and 2 both raised concerns over whether the perturbations made for the storylines of precondition severity would violate any correlation structures between PET and temperature (hence the choice of PET on the y-axis). We believe this to be appropriate given the fact that PET and precipitation are both inputs to the GR4J hydrological model and this comparison directly shows that the perturbations made to the input data to the hydrological model are physically plausible.

Fig. 4: Please use another encoding than colour for the points on the map (e.g. varying point shapes).

Response: Done.

Fig.7: Same color gradient for latitude and persistence time is confusing. Perstence time could be binned and log-scaled. Please give a metric for the panels in Fig. 7 that quantifies the strength of the relationship between x and y.

Response: Done.

Regarding the conclusions: Please state clearly here what is found not what is done (e.g. sentences L693-699 are too vague). Paragraph L 711-715 is not really helpful, what is the value/the message here? Please consider that storyline approaches are relatively new for the community and the benefits compared to climate change scenario modelling approaches should be a) clearly mentioned and b) justified by some values.

Response: Lines 711-715 places this study in relation to practical water resources planning and the potential to bridge probabilistic (return period) estimates with the storyline approach to understand extreme droughts with no historical precedent. We agree it might be better for this to be placed elsewhere and a shortened version of this has been included in the discussion (lines 564-567).

We have rewritten the conclusions to show 1) the extent to which the 2010-12 drought could have been worse given the various storylines and 2) that the storyline approach adds value and complements traditional climate change impact assessments (lines 621-644). To further clarify the added value of the storyline approach compared to existing approaches, we have also added reference to a recently accepted review paper (Chan et al. in press) (lines 56-61 and lines 561-564). The review paper categorizes studies on the hydrological impacts of climate change in the UK by their methodological approaches and outlines outstanding research gaps that can be tackled by emerging approaches such as the storyline approach.

**References**

Chan, W.C.H., Shepherd T.G., Facer-Childs, K., Darch, G. and Arnell, N.: Tracking the methodological development of climate change projections for UK river flows, Progress in Physical Geography: Earth and Environment. in press.

Shepherd, T. G.: Bringing physical reasoning into statistical practice in climate-change science, Climatic Change, 169, 2, https://doi.org/10.1007/s10584-021-03226-6, 2021.

Wilby, R. L. and Harris, I.: A framework for assessing uncertainties in climate change impacts: Low-flow scenarios for the River Thames, UK, Water Resources Research, 42(2), https://doi.org/10.1029/2005WR004065, 2006.

Zscheischler, J., Martius, O., Westra, S., Bevacqua, E., Raymond, C., Horton, R. M., van den Hurk, B., AghaKouchak, A., Jézéquel, A., Mahecha, M. D., Maraun, D., Ramos, A. M., Ridder, N. N., Thiery, W., and Vignotto, E.: A typology of compound weather and climate events, Nat Rev Earth Environ, 1, 333–347, https://doi.org/10.1038/s43017-020-0060-z, 2020.

---

## Author Response (AR4)

Dear Jim,

Thank you for this positive decision in accepting our manuscript for publication in HESS. I am grateful for the comments and suggestions by yourself and the reviewers which have improved our analyses and clarified core arguments in the paper. We appreciate your time and effort in coordinating and handling the review process of the manuscript.

Please find attached the production files for publication.

Thank you again.

Best wishes,
Wilson